# Taotie neurons regulate appetite in *Drosophila*

Yin Peng Zhan[1,2], Li Liu[1,2] & Yan Zhu[1,2]

The brain has an essential role in maintaining a balance between energy intake and expenditure of the body. Deciphering the processes underlying the decision-making for timely feeding of appropriate amounts may improve our understanding of physiological and psychological disorders related to feeding control. Here, we identify a group of appetite-enhancing neurons in a behavioural screen for flies with increased appetite. Manipulating the activity of these neurons, which we name Taotie neurons, induces bidirectional changes in feeding motivation. Long-term stimulation of Taotie neurons results in flies with highly obese phenotypes. Furthermore, we show that the *in vivo* activity of Taotie neurons in the neuroendocrine region reflects the hunger/satiety states of un-manipulated animals, and that appetitive-enhancing Taotie neurons control the secretion of insulin, a known regulator of feeding behaviour. Thus, our study reveals a new set of neurons regulating feeding behaviour in the high brain regions that represents physiological hunger states and control feeding behaviour in *Drosophila*.

[1] State Key Laboratory of Brain and Cognitive Science, Institute of Biophysics, Chinese Academy of Sciences, Beijing 100101, China. [2] University of the Chinese Academy of Sciences, Beijing 100049, China. Correspondence and requests for materials should be addressed to Y.Z. (email: zhuyan@ibp.ac.cn).

Feeding is essential to generate the energy required to support life in all its forms, ranging from unicellular bacteria to multicellular mammals[1,2]. To maintain energy homeostasis, the brain of most higher animals monitors changes in their internal physiological states, and responds with corresponding behavioural changes in feeding actions[3]. To understand how this process occurs at the neural circuit level is of fundamental importance, as it not only establishes a general model for state-dependent modification of behaviour, but also provides clues to neural mechanisms of feeding-related psychological and physiological disorders[4]. For instance, abnormally increased appetite causes elevated food intake, regardless of physiological needs, and usually leads to a state of obesity[5].

*Drosophila melanogaster* represents an excellent model for investigating the essential molecular and neuronal mechanisms underlying feeding behaviour, for both wide availability of powerful genetic tools and highly conserved molecular pathways from flies to mammals in metabolic homeostasis[6]. Even in a seemingly simple organism such as the fruit fly, feeding is a complex process that involves motivational, sensory and motor circuits[7–9]. Significant progress has been made in our understanding of the gustatory inputs for assessing the quality of food[10], and in identifying the feeding command neurons for motor actions[11,12]. However, the organization of feeding control in the central brain at the circuitry level remains poorly understood. Several neurotransmitters and neuropeptides have been implicated in mediating the processes of feeding behaviour in *Drosophila*[13–15]. For example, studies of the dopamine and its receptor neurons have revealed that distinct dopaminergic neurons in different brain areas act at multiple control points[15–18]. Recently, four GABAergic interneurons were identified to be involved in gating motor neurons in feeding initiation and ingestion[19]. While the precise nature of the neuronal substrates remains to be elucidated, much of the signalling system appears to be functionally conserved throughout evolution[20]. Interestingly, part of the serotonergic network, which projects broadly into the brain, was recently implicated in both inducing feeding and promoting appetitive memory performance[21].

Similar to mammals, feeding behaviour in *Drosophila* is regulated not only by signals within the brain, but also by endocrine signalling, which relays physiological status to the brain to influence feeding decisions. The circulating hormones from the neuroendocrine system and fat body (analogous to adipose tissues in vertebrate), such as insulin-like peptides (DILPs) and adipokinetic hormone (AKH), modulate feeding behaviour by representing the internal energy status[22,23]. Additionally, specialized central brain neurons directly sense certain circulating nutrients and modify feeding behaviour[24,25].

The complexity of feeding behaviour and intertwining neural-physiological processes renders the task of understanding the neural controls of feeding behaviour difficult. The motivational feeding plays a critical role in overfeeding that leads to overweight and obesity. Despite this is a common feature of feeding decision in all animal species, how the brain calculates and represents the physiological states of hunger and satiety for feeding controls and energy homeostasis has remained elusive.

Here, using behavioural analyses, genetic manipulations as well as neural imaging, we reveal a group of neurons that are a key neural substrate for the hunger and satiety response in flies, both neurologically representing the physiological hunger state and gating feeding behaviour in the *Drosophila* brain.

## Results

### Activating Taotie neurons evokes vigorous feeding behaviour.
As for most other animals, fully fed (satiated) flies do not eat when presented with food, whereas starved (hungry) flies ingest an abundance of food[2,26]. In order to gain novel insights into neural circuits integrating feeding controls with physiological needs in *Drosophila*, we designed a behaviour-based appetite screen for appetite-promoting neurons. We speculated that in satiated flies, once the neurons responsible for hunger-elicited feeding are artificially activated, they will increase the motivation for feeding, that is the flies would appear to have elevated appetite (Fig. 1a). Using dTrpA1 (ref. 27) as a thermogenetic activator, we screened approximately 400 GAL4 lines known to be expressed in adult *Drosophila* brains using the amount of food intake as readout. Out of those 400 lines, one line was identified, which we named *Taotie-GAL4*, with the corresponding neurons termed *Taotie* (after an ancient gluttonous ogre from Chinese mythology). In satiated flies, activation of Taotie neurons with *dTrpA1* at 30 °C promptly evoked vigorous feeding (Fig. 1b).

To quantify the stimulated feeding behaviour of these flies with elevated appetite, we first classified flies according to the amount of dye-stained food in their gut. Although little feeding occurred in controls, over half of *Taotie > TrpA1* flies exhibited strong feeding behaviour at increased temperature (Fig. 1c). Furthermore, using colorimetric quantification of the dye in fly homogenates, we found that on average, activated *Taotie-GAL4* rapidly increased food consumption over fivefold relative to the control in the same test period (Fig. 1d). Interestingly, in hungry flies (starved for over 60 h), no difference in food intake was found between the activation group and the control group (Supplementary Fig. 1a). The lack of synergistic interaction between activation of Taotie neurons and starvation treatment suggests that elevated appetite due to activation of Taotie neurons is part of the hunger response. Besides *dTrpA1*, activating *Taotie-GAL4* with an optogenetic activator *CsChrimson*[28] in satiated flies also promoted the intake of much higher volumes of food than that in controls in a light-dependent manner (Fig. 1e). Together, our results suggest that hyperactivating Taotie neurons in satiated flies results in a strongly orexigenic phenotype, and overcomes suppressive satiety mechanisms.

We next tested whether forcible activation of *Taotie-GAL4* evokes the motivation for feeding. First, following activation of Taotie neurons, the satiated flies displayed higher rates of proboscis extension reflex (PER) for sugar solutions of much lower concentrations compared to the controls in PER assay[2] (Fig. 1f and Supplementary Fig. 1b). Notably, like the starved flies, these flies did not exhibit spontaneous PER in the absence of food. This context-dependent feeding action was distinct from that elicited by activating neurons commanding motor actions of feeding[12] (Fig. 1g). In addition, similar to starved flies, the satiated flies with activated Taotie neurons were attracted by a food odour, apple cider vinegar, clearly displaying signs of hunger (Fig. 1h,i). In summary, these data demonstrate that forced activation of Taotie neurons raises the motivation of satiated flies, reaching a state that is typically induced by hunger.

### Reversible obesity by chronic Taotie activation.
To evaluate activation of Taotie neurons as a *Drosophila* model for eating disorders, we next examined the long-term effects of chronically increased appetite in satiated flies. Throughout the following experiments, the *Taotie > CsChrimson* and control flies were allowed to eat *ad libitum*. After Taotie neurons were continuously activated for 4 days by optogenetics, the probability of PER was significantly increased in *Taotie > CsChrimson* flies (Fig. 2a), even though the abdomen of these flies contained over two-fold more food compared with controls (Fig. 2b), suggesting an elevated feeding motivation in chronically activated flies.

Under normal conditions, feeding increases the levels of glucose and triglyceride in the blood, which conversely act as satiation signal to inhibit feeding, thus maintaining the energy homeostasis of the whole body[29]. We therefore investigated the long-term effects of chronic activation of Taotie neurons. As shown in Fig. 2c, after 4 days of persistent

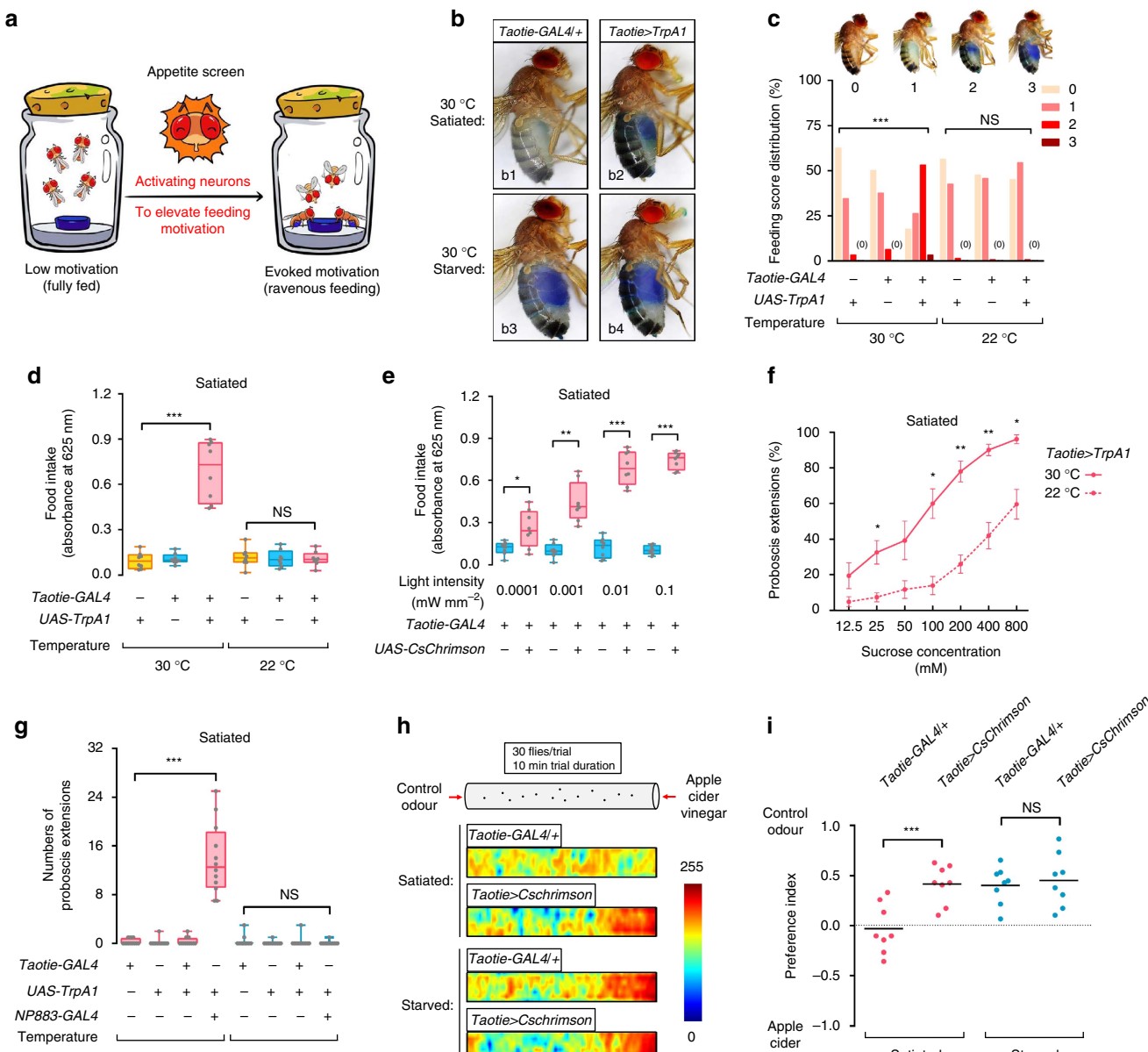

**Figure 1 | Activation of Taotie neurons in satiated flies evokes voracious feeding behaviour and increases feeding motivation. (a)** A cartoon presentation of the 'appetite screen' experiment. Normally, satiated flies rarely eat food (left vial); in contrast, forcibly activating specific neurons in the brain could elevate 'appetite' and drive satiated flies to ingest an abundance of food (right vial). **(b)** Dye-stained food in the abdomen of *Taotie-GAL4/UAS-TrpA1* flies when activating Taotie neurons at 30 °C (b2 and b4). The negative control was satiated *Taotie-GAL4/ +* flies (b1), the positive control was starved *Taotie-GAL4/ +* flies (b3). **(c)** Distribution of feeding scores of satiated *Taotie-GAL4/UAS-TrpA1* flies at 30 °C and 22 °C. Zero indicates none of the flies showed a phenotype in the corresponding category (n = 120 flies per condition). **(d)** Colorimetric quantification of food-intake in *Taotie-GAL4/UAS-TrpA1* satiated flies (N = 8 groups per condition, n = 20 flies in each group). **(e)** Food ingested by satiated *Taotie-GAL4/UAS-CsChrimson* flies under optogenetic stimulation with different light intensities (N = 8). **(f)** Fractions of satiated *Taotie-GAL4/UAS-TrpA1* flies showing PER to different concentrations of sucrose (N = 4–5, n = 5–13). **(g)** In the absence of food, spontaneous proboscis extension was observed in *NP883-GAL4/UAS-TrpA1* flies at 30 °C, but not in *Taotie-GAL4/UAS-TrpA1* flies (N = 12). *NP883* contains Fdg neurons, the motor command neurons for feeding. **(h)** Heat-maps of representative food odour attraction in different flies. In each tube, the left end was delivered with water vapour, while the right end was delivered with vapour of apple cider vinegar. Satiated *Taotie-GAL4/ +* flies served as a negative control and starved *Taotie-GAL4/ +* flies as a positive control. **(i)** Comparison of food odour attraction behaviours of *Taotie-GAL4/UAS-CsChrimson* flies and controls (N = 8). All genotypes, temperatures and experimental conditions are indicated with the plots. In a box and whisker plot, whiskers mark minimum and maximum, box includes 25th – 75th percentile, and the line in box indicates median of the data set. NS indicates not significant (P > 0.05); *P < 0.05, **P < 0.01, ***P < 0.001 (Student's t-test for two-group only comparisons, ANOVA with Bonferroni *post hoc* test for multiple comparisons). *Fisher's exact* test for (**c**) (compare all three groups at 30 °C or 32 °C). Error bars indicate s.e.m.

activation, the body weight of *Taotie > CsChrimson* flies increased significantly. In addition, the levels of triglyceride and glucose were increased as well (Fig. 2d and Supplementary Fig. 2a). Furthermore, chronic activation of *Taotie-GAL4* caused a massive accumulation of body fat under the cuticle (Fig. 2e). Together, these data demonstrate that artificially activating Taotie neurons forces the energy homeostasis toward a more obese phenotype,

irrespective of the opposing influence of a physiologically satiated body.

In human, obesity is usually an irreversible condition, especially in the long term[4,5]. In contrast, all of the obese phenotypes described above reverted back to their normal states within 4 days of no activation. Importantly, this process was accompanied by a reduction of food intake lower than baseline

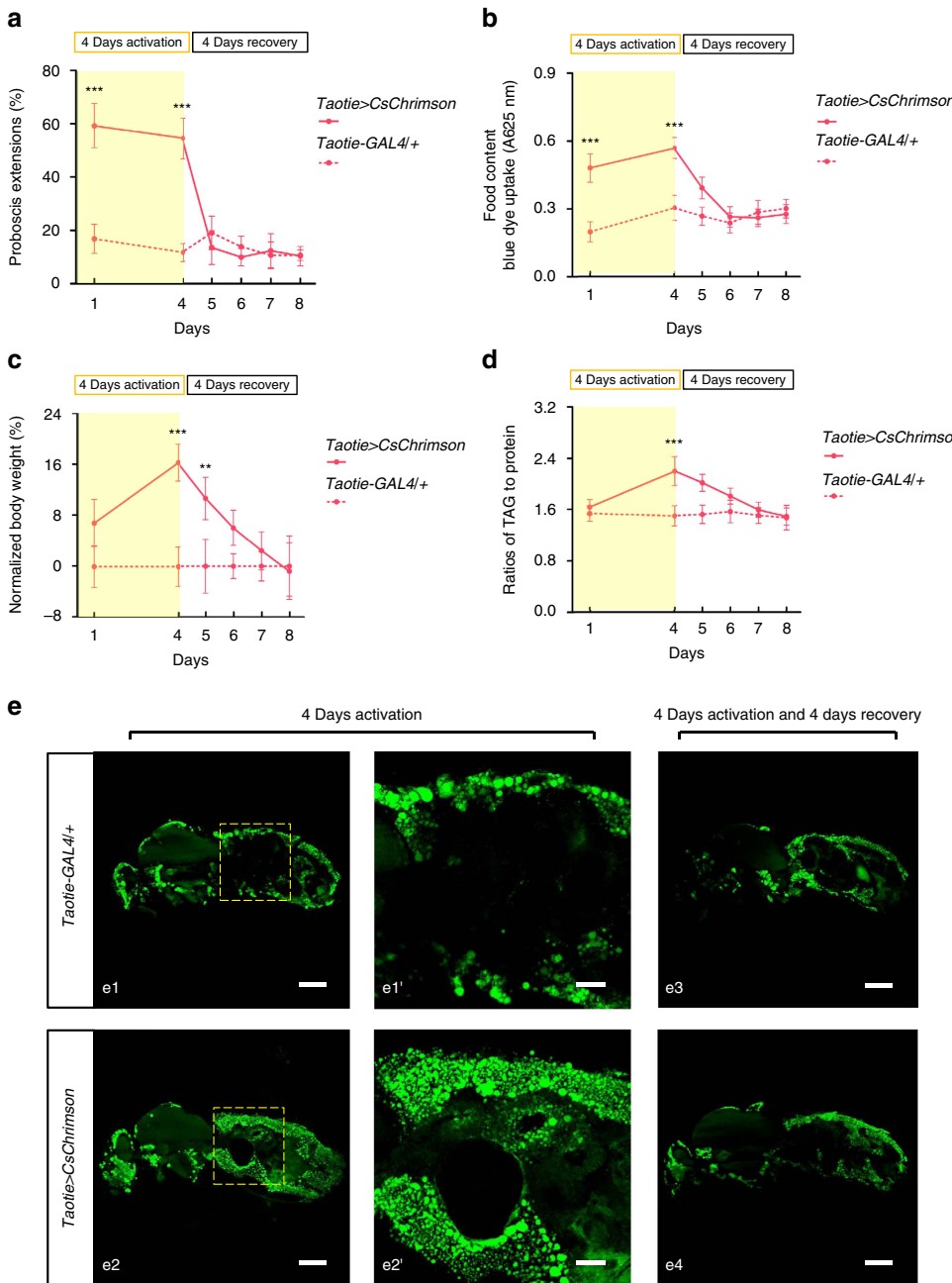

**Figure 2 | Chronic activation of Taotie neurons induces reversible obese phenotypes.** *Taotie-GAL4/UAS-CsChrimson* and genetic control flies were activated by continuous exposure to light for 4 days, and then allowed to recover for another 4 days without light. Flies had *ad libitum* access to food during the entire time. The following tests were conducted at two time points: (1) after 4 days of activation, and (2) after 4 days of recovery. (**a**) Fraction of satiated flies, *Taotie-GAL4/UAS-CsChrimson* and genetic control, showing PER to 100 mM sucrose ($N = 4$–5, $n = 6$–10). (**b**) Food content in *Taotie-GAL4/UAS-CsChrimson* flies ($N = 8$). (**c**) Comparison of body weight of *Taotie-GAL4/UAS-CsChrimson* and genetic control flies ($N = 6$). Changes of body weight were normalized to *UAS-CsChrimson/+* control flies. (**d**) Triglyceride levels, normalized to protein levels, in *Taotie-GAL4/UAS-CsChrimson* and genetic control flies ($N = 6$). (**e**) Cryo-sections showing lipid distributions in female flies after activation or recovery. e1–e4: Low magnification images showing the whole body lipid distribution. Scale bar, 300 μm. (e1′, e2′) High-magnification images of the boxed regions in e1 and e2 showing the sub-cuticular lipid in abdominal regions. Scale bar, 100 μm. All genotypes and experimental conditions are indicated with the plots. NS indicates not significant ($P > 0.05$); **$P < 0.01$, ***$P < 0.001$ (Student's *t*-test for two-group comparisons). Error bars indicate s.e.m.

levels after cessation of light stimulation (Fig. 2a–e and Supplementary Fig. 2a,b). The existence of such reversible obese phenotypes strongly suggests that once the commanding neurons shift the energy requirement to a different set point, the body is physiologically flexible to accommodate the necessary changes in either direction.

Because an increase in body fat alters the body's response to insulin, potentially leading to disrupt insulin-related signals in mammals[30], we measured insulin-related signals in the induced obese flies. Interestingly, transcription levels of *Dilp2*, *Dilp3* and *Dilp5* increased upon chronic activation of Taotie neurons whereas downstream signals of insulin receptor were impaired in these flies, suggesting that chronic activation of Taotie neurons impairs insulin-related signals (Supplementary Fig. 3a,b). In mammals, obesity-associated inflammation is an important contributor to the decrease of insulin-related signals[31,32]. Similarly, in flies with chronically activated Taotie neurons, the transcriptional levels of *Eiger* and *Upd3*, homologues of mammalian TNF and inflammatory cytokine[33,34], were found to be considerably higher compared to control, suggesting a common inflammatory response in Taotie neuron-induced obese flies and in obese mammals (Supplementary Fig. 3c). Fat-body-expressed cytokine unpaired (*Upd2*) acts as a functional homologue of mammalian leptin, which transfers satiated signals from adipose tissue to the central neural system[35]. In flies with chronic activation of Taotie neurons, the transcription level of *Upd2* was lower compared with control, suggesting a cross-talk mechanism between Taotie neurons and metabolic tissues (Supplementary Fig. 3d).

To gain further insights into the mechanisms that induced obesity following activation of Taotie neurons, we challenged *Taotie > CsChrimson* flies with a high sugar diet. These flies exhibited a dramatic elevation of the levels of triglyceride and glucose under this treatment, suggesting that energy homeostasis in the flies with activated Taotie neurons were further skewed under the challenge of high energy diet (Supplementary Fig. 2c,d). To further characterize the relationship of obese phenotypes and the activity level of Taotie neurons, we took advantage of optogenetic stimulation to activate Taotie neurons with different light intensities, which generated distinct degrees of obese phenotypes, suggesting that the activity levels of Taotie neurons function to fine-tune the energy homeostasis in flies (Supplementary Fig. 2e,f).

**Feeding motivation requires Taotie neurons**. To examine the role of Taotie neurons in the hunger response of the brain, we asked whether inhibition of Taotie neurons suppresses food intake in hungry flies. We used tetanus toxin light chain (TNT)[36] together with a temperature-sensitive GAL80 (*Tub-Gal80ts*)[37] to selectively inhibit synaptic transmission of targeted neurons in pre-defined time windows. As shown in Fig. 3b, suppressing Taotie neurons strongly reduced food intake in hungry flies (starved for 24 h). Although satiated flies displayed very weak feeding responses, their feeding was further reduced by inactivation of *Taotie-GAL4* (Fig. 3a). Furthermore, inactivation of *Taotie-GAL4* resulted in a lower probability of PER in both starved and satiated flies (Fig. 3c and Supplementary Fig. 1c,d). Therefore, blocking Taotie neurons renders flies to reach satiation prematurely, despite the fact that physiologically, these starved flies remain hungry.

We wondered whether the phenotype of anorexia results from loss of gustatory discrimination or inadequate motor controls. Similar to genetic controls, the starved flies with blocked Taotie neurons preferred higher concentration sucrose in the discrimination test (Fig. 3d). This is distinct from blocking the

sugar sensing neurons, Gr64f[38], supporting the idea that the anorexia following *Taotie-GAL4* inactivation is not due to inability of detecting food. The results of the food discrimination test, together with the previous PER experiment, indicate that inactivation of *Taotie-GAL4* does not block the motor programme of feeding. Considering Taotie neurons are crucial for regulating feeding motivation and energy homeostasis, we next asked whether blocking the activity of Taotie neurons influence physiological states in these flies. As shown in Fig. 3e,f, inactivation of Taotie neurons effectively attenuated the elevation of glucose and triglyceride levels by high sugar diet, but not by regular food, suggesting that activity of Taotie neurons is important for bidirectional control of feeding behaviour and physiological homeostasis. Taken together, the activation and inactivation experiments demonstrate that the proper activity of Taotie neurons are both necessary and sufficient for controlling feeding motivation and regulating energy homeostasis in flies.

**Activating Taotie neurons represents a hungry state**. Previous studies revealed that fruit flies adopt distinct homeostatic strategies to coordinate feeding behaviour in response to different internal hunger states[8,16,39]. To explore the relationship between the evoked feeding following *Taotie-GAL4* activation and hunger-induced feeding, we characterized the temporal features of feeding behaviour of these flies. We found that the amount of food consumed by flies upon *Taotie-GAL4* activation was similar to that of flies starved for 12 h (Fig. 4a,b and Supplementary Fig. 1e). Furthermore, to evaluate the effects of starvation on subsequent feeding, flies were starved for various durations before measuring food intake. As shown in Fig. 4c, with increasing length of starvation, the control flies gradually increased their food intake, indicating a gradually arousing hunger drive. In contrast, the flies with Taotie neurons activated after starvation for 4–12 h maintained a relatively stable level of feeding (Fig. 4c). As a result, the difference between the activated group and the starved group became progressively smaller with increased starvation time, suggesting that *Taotie-GAL4* activation mimics the effect of 12 h starvation (Fig. 4c and Supplementary Fig. 1f). Together, these results indicate that activation of Taotie neurons is an integral event of a hunger response.

Because feeding of hungry flies is primarily influenced by the quality of food (nutrition content and palatability)[2,40], we asked whether activation of Taotie neurons results in similar food preferences. As shown in Fig. 4d, satiated flies with activated Taotie neurons fed vigorously, but not indiscriminately: they still exhibited a distinct preference toward palatable sugars and proteins-rich foods, but not toward tasteless sugar, despite its nutritional value (Supplementary Fig. 4). More importantly, their food preferences were similar to those of hungry flies (food-deprived for 12 h) (Fig. 4e). Furthermore, those activated flies and hungry flies exhibited similar responses toward various mixtures of arabinose (non-nutritious yet palatable) and sorbitol (tasteless but nutritious) (Supplementary Fig. 5a–f). Lastly, food containing bitter compounds curbed the feeding in both fly types, but not in severely starved flies (Supplementary Fig. 5g–i). Together, with different classes of food tested, *Taotie-GAL4* activation evokes an identical pattern of food preference as starvation, suggesting that activation of *Taotie-GAL4* is equivalent to stimulation of appetite state in flies, in other words, these flies are neurologically hungry, even though they are physiologically satiated.

**Taotie activation induces persisting hunger**. In order to test whether activation of Taotie neurons generates an enduring arousal state beyond the time of activation, we measured the temporal effect of transient activation on feeding in satiated flies

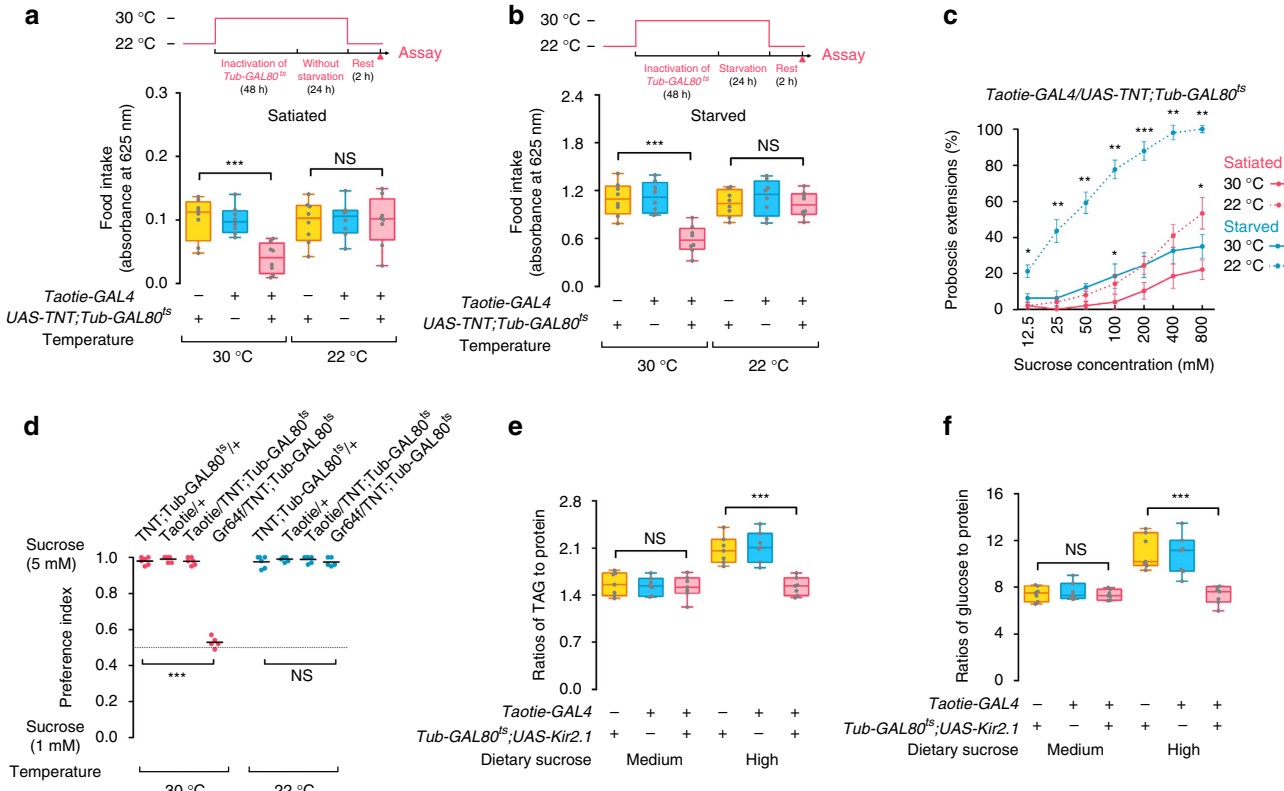

**Figure 3 | Inactivation of Taotie neurons reduces feeding motivation in satiated and starved flies.** (**a,b**) Amount of food ingested in *Taotie-GAL4/UAS-TNT;Tub-GAL80ts* flies in satiated (**a**) and starved (**b**) conditions (N = 8). Prior the feeding tests, flies in the 30 °C group were treated for 2 days at 30 °C for heat-inducible expression of TNT. Control flies (22 °C, non-induced) were treated in identical manner, except that the temperature was 22 °C (see also *Materials and Methods*). (**c**) Fraction of *Taotie-GAL4/UAS-TNT;Tub-GAL80ts* flies displaying PER to different concentrations of sucrose in satiated or starved conditions (N = 4–5, n = 7–13). (**d**) Preference index of starved flies in a sucrose preference assay. *Gr64f-GAL4/UAS-TNT;Tub-GAL80ts* flies served as a negative control (N = 5–6). (**e,f**) Levels of triglyceride and glucose, normalized to protein levels, in *Taotie-GAL4/Tub-GAL80ts;UAS-Kir2.1* flies and genetic control flies on regular food and high sugar diet (N = 7). All genotypes, temperatures and experimental conditions were as indicated with the plots. In a box and whisker plot, whiskers mark minimum and maximum, box includes 25th – 75th percentile, and the line in box indicates median of the data set. NS indicates not significant (P > 0.05); *P < 0.05, **P < 0.01, ***P < 0.001 (Student's *t*-test for two group-only comparisons, ANOVA with Bonferroni *post hoc* test for multiple comparisons). Error bars indicate s.e.m.

using optogenetics. The satiated *Taotie > CsChrimson* flies first received light stimulation for a minimum of 30 min in the absence of food. The flies were then moved into and kept in darkness before the start of the subsequent feeding assay. Following this procedure, *Taotie > CsChrimson* flies exhibited strong food intake, with no further activation required (Fig. 5a and Supplementary Fig. 6a). In contrast, such prolonged effect was absent when we tested several gustatory receptors[41] and the motor command neurons for feeding actions[12] with the same procedure (Fig. 5b). As typically sensory neurons and motor command neurons have fast response dynamic, the temporal effect of Taotie activation strongly suggests that these neurons are not located in the input or output pathways of feeding control, but have a central position in the command chain for feeding decisions.

To determine for how long this after-effect is sustained, satiated flies of *Taotie > CsChrimson* were activated by light only once, then kept in the dark without food for varying durations, prior to starting the feeding assay (also conducted in darkness). As shown in Fig. 5c, after transient activation of *Taotie-GAL4*, the flies exhibited no decay in feeding motivation for the next 4 h, without further light activation or food stimulation during this period (Supplementary Fig. 6b). Besides food intake, the probability of PER showed a similar temporal trend after

brief activation of Taotie neurons, suggesting transient *Taotie-GAL4* activation elicits a long-lasting motivation for feeding (Fig. 5d). Furthermore, we also observed similar persistent after-effects when activating *Taotie-GAL4* with thermogenetics (Supplementary Fig. 6c–f). Interestingly, providing flies with food in the delay period suppressed the long-lasting feeding motivation, indicating that satiety signals generated from feeding overcome the prolonged feeding motivation in the absence of continuous activation of Taotie neurons (Supplementary Fig. 7). Together, these results demonstrated that transient activation of Taotie neurons forces these flies to enter into a long-lasting aroused state with elevated motivation for feeding, which further suggests that the activity of these neurons defines the hunger state in the brain, rather than the act of feeding itself.

**Taotie neurons in PI region encode hunger state.** We next investigated the expression pattern of *Taotie-GAL4*. The *Taotie > GFP*-labelled neurons were found scattered throughout the brain (in the sub-esophageal zone (SEZ), *pars intercerebralis* (PI), dorsal and lateral protocerebrum, and ellipsoid body) and in the ventral nerve cord (VNC) (Fig. 6a). *Taotie-GAL4* also labelled neurons in the periphery: the proboscis and tarsi of the legs; nevertheless, their involvement in feeding was ruled out by genetic and surgical manipulation (Supplementary Fig. 8).

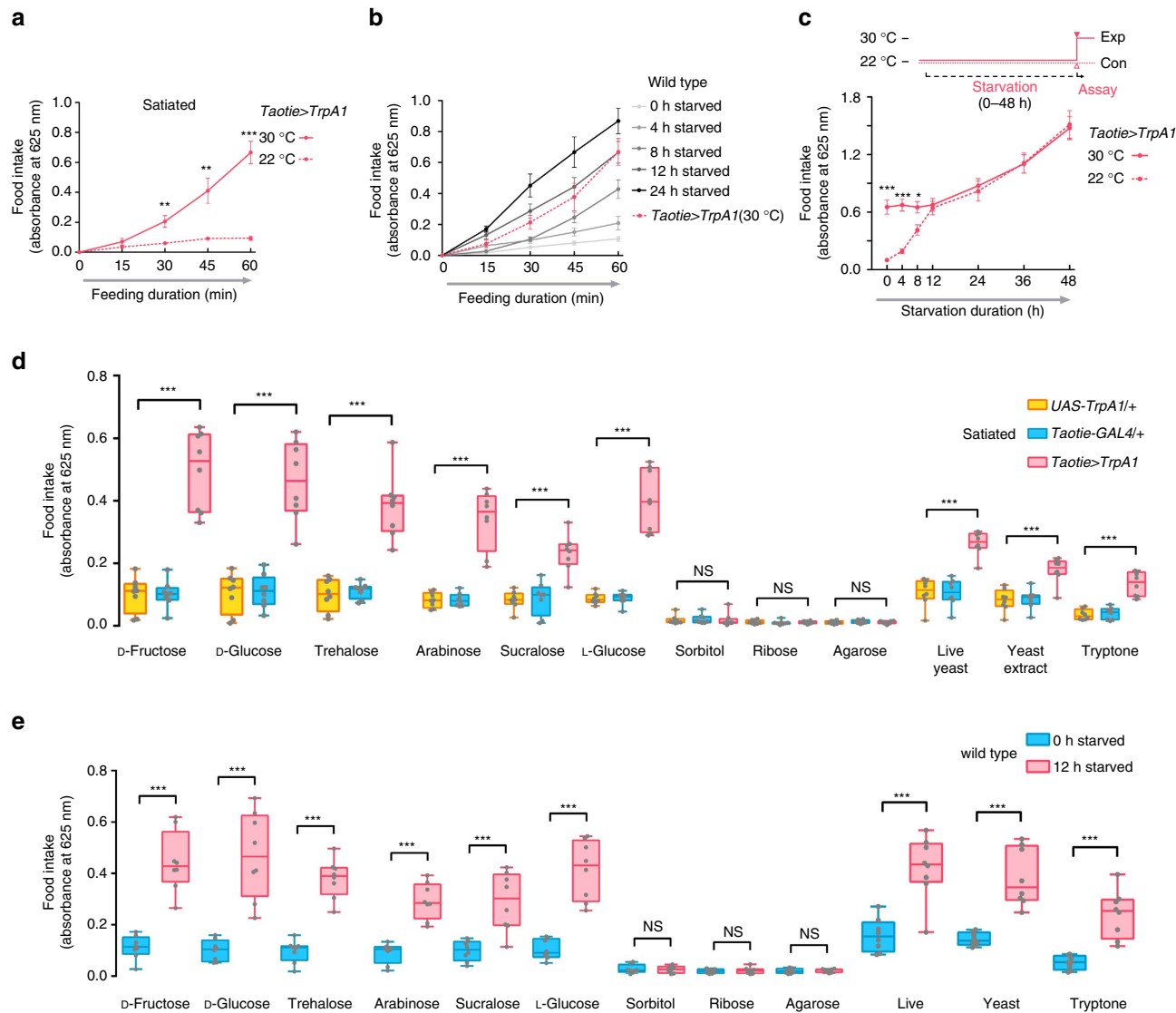

**Figure 4 | Satiated flies with activated Taotie exhibit identical behavioural patterns as hungry flies.** (**a**) The quantities of food consumed by satiated *Taotie-GAL4/UAS-TrpA1* flies through 1 h feeding assay (*N* = 8). (**b**) The quantities of food consumed by wild type flies after different starvation durations (0–24 h) (*N* = 8). For comparison, the food intake of satiated *Taotie-GAL4/UAS-TrpA1* flies at 30 °C was plotted again as a red dotted line. (**c**) Performance of *Taotie-GAL4/UAS-TrpA1* flies with different durations of starvation at 22 °C before activation of Taotie neurons for feeding assay (*N* = 8). All temperature control flies were reared and starved at 22 °C, with feeding assays performed at the same temperature. (**d**) Food intake of satiated *Taotie-GAL4/UAS-TrpA1* and genetic control flies when given different carbohydrates or protein-rich diets (5% w/v) at 30 °C (*N* = 8). Sugars tested belong to four categories: (1) nutritious and palatable carbohydrates (D-fructose, D-glucose and trehalose), (2) non-nutritious yet palatable carbohydrates (arabinose, sucralose and L-glucose), (3) tasteless but nutritious carbohydrate (sorbitol), and (4) tasteless and non-nutritious (ribose and agarose). All sugars were used at 200 mM, except sucrose and sucralose at 100 mM, to provide equal nutritional values for mono- and di-saccharides. Live yeast, yeast extract and tryptone served as protein-rich food sources. (**e**) Wild type flies starved for 12 h were tested with different carbohydrates or protein-rich diets similar to (**d**) (*N* = 8). All genotypes, temperatures and experimental conditions are indicated with the plots. In a box and whisker plot, whiskers mark minimum and maximum, box includes 25th – 75th percentile, and the line in box indicates median of the data set. NS indicates not significant (*P* > 0.05); *$P$ < 0.05, **$P$ < 0.01, ***$P$ < 0.001 (Student's *t*-test for two group-only comparisons, ANOVA with Bonferroni *post hoc* test for multiple comparisons). Error bars indicate s.e.m.

In addition, this observation was supported by laser targeting activation experiments. There, focusing a laser beam on the head, instead of thorax, was sufficient to evoke PER behaviour toward sucrose in satiated *Taotie* > *CsChrimson* flies (Supplementary Fig. 9). Therefore, the activity of Taotie neurons in the central brain is sufficient for feeding motivation.

To determine the functional subset of Taotie neurons in the central brain, we surveyed GAL80 transgenes to restrict the expression of *Taotie-GAL4*. Except *Elav-GAL80*, none of the

individual GAL80s or their combinations reverted the overeating phenotype. A remaining mini-population, generated by combining *Gad-GAL80* and *Cha-GAL80*, was still sufficient to evoke hyper-feeding behaviour in satiated flies (Fig. 6b and Supplementary Fig. 10). This remaining subset includes one cluster of PI neurons and two neurons in the lateral protocerebrum, with additional 6–7 cells scattered irregularly in the dorsal region (Fig. 6c). Labelling with pre-synaptic marker and post-synaptic marker revealed that Taotie neurons receive

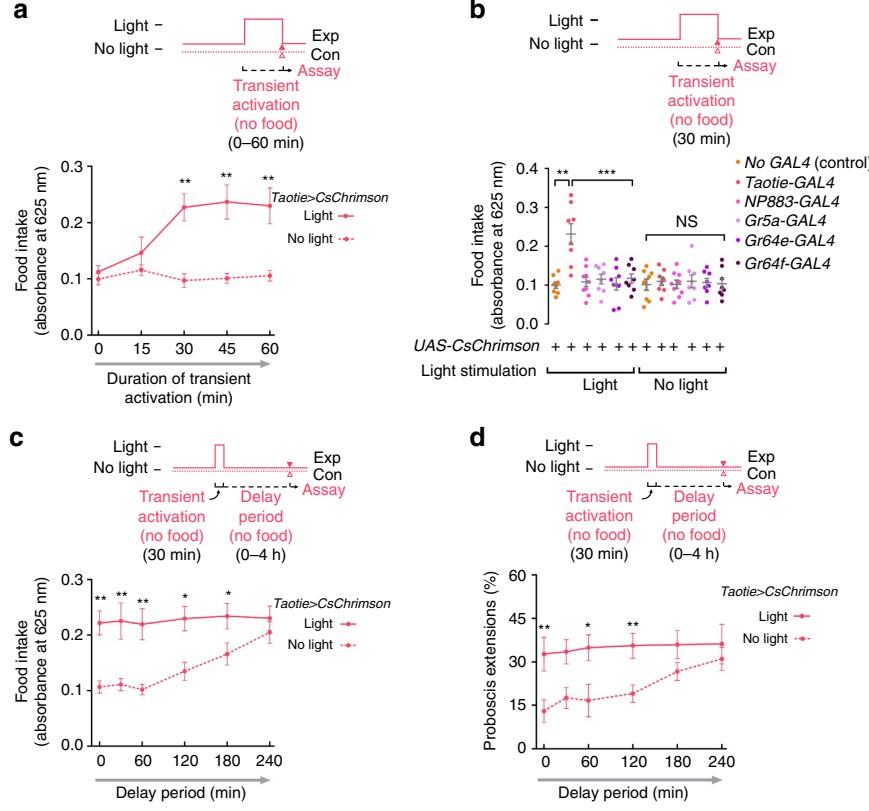

**Figure 5 | Transient activation of Taotie neurons prolongs feeding motivation.** (**a**) Satiated flies of *Taotie-GAL4/UAS-CsChrimson* were transiently activated by orange light for 0–60 min, and then immediately tested in the dark for food-intake assay (N = 8). 'No light' flies were put through the same procedure, but without light stimulation. (**b**) Comparison of food intake of satiated flies following activation of known sweet gustatory receptors (Gr5a, Gr64e and Gr64f) and putative feeding command neurons (*NP883-GAL4*) by optogenetics, and prior to subjecting flies to our feeding assay (N = 8). (**c**) Food intake of *Taotie-GAL4/UAS-CsChrimson* flies during various time intervals between activation and feeding test (N = 8). During the intervals, the flies were kept in the dark without food. 'No light' flies were exposed to the same procedure without light stimulation. (**d**) After-effect of PER behaviour in *Taotie-GAL4/UAS-CsChrimson* flies with different intervals between transient activation and PER test (N = 5–6, n = 5–15). Flies were maintained in the dark without food during these intervals. All genotypes, temperatures and experimental conditions were as indicated with the plots. NS indicates not significant (P > 0.05); *P < 0.05, **P < 0.01, ***P < 0.001 (Student's *t*-test for two group-only comparisons, ANOVA with Bonferroni *post hoc* test for multiple comparisons). Error bars indicate s.e.m.

multiple inputs and then project into the sub-esophageal zone, thus directly modulating feeding behaviour (Supplementary Fig. 11). Furthermore, specifically silencing these neurons at the adult stage significantly inhibited feeding behaviour in starved flies (Fig. 6d).

Next, we sought to identify intracellular signals pathways required by Taotie neurons for feeding promotion. While the promoter of *Taotie-GAL4* contained the upstream sequence of gustatory receptor *Gr28b.b*, deletion or overexpression of *Gr28b.b* gene had no effect on the amount of food intake, suggesting the receptor itself is not involved in Taotie-evoked feeding behaviour (Supplementary Fig. 12). Another clue came from a proprotein convertase, encoded by *amontillado* (*amon*), which is required for processing precursors into neuropeptides before their release from neurosecretory cells[42]. Interestingly, when we knocked down *amon* in *Taotie > ReaChR* flies, overfeeding behaviour evoked by Taotie neuron activation was suppressed (Supplementary Fig. 13a). Furthermore, overexpression of *amon* in Taotie neurons increased the food content in adults, while knocking down of *amon* in Taotie neurons inhibited food intake in starved flies (Supplementary Fig. 13b,c). These results strongly suggest that Taotie neurons evoke feeding behaviour in a neuropeptide-dependent manner. Together, our results indicate

that manipulation of a small population of neurons in the neurosecretory area of the high brain is necessary and sufficient to evoke feeding in satiated flies and suppress feeding in hungry flies.

We next analysed the relationship between neural activity of this mini-population and the hunger/satiety states of non-manipulated flies. As shown in Fig. 6e, compared to satiated flies, the neurons in the PI region exhibited increased GCaMP fluorescence following starvation, indicating that, when flies are hungry, these neurons are activated hence have high levels of intracellular calcium. Interestingly, feeding hungry flies for 1 h reduced the calcium level in these neurons to the level of satiated flies (Fig. 6e,f). In contrast to the PI region, no changes of calcium level were detected in the pair of cells in the lateral protocerebrum, regardless of whether the flies were hungry or not (Supplementary Fig. 14a). Together, our results strongly suggest that the activity of Taotie neurons encodes the physiological states of hunger and satiety in flies.

To establish the causal link between physiological factors and activity of Taotie neurons, we treated flies with sugars of different nutritional values, and monitored neural activity of Taotie neurons using GCaMP. We fed two groups of flies with either D-glucose (nutritious and palatable) or L-glucose (non-nutritious

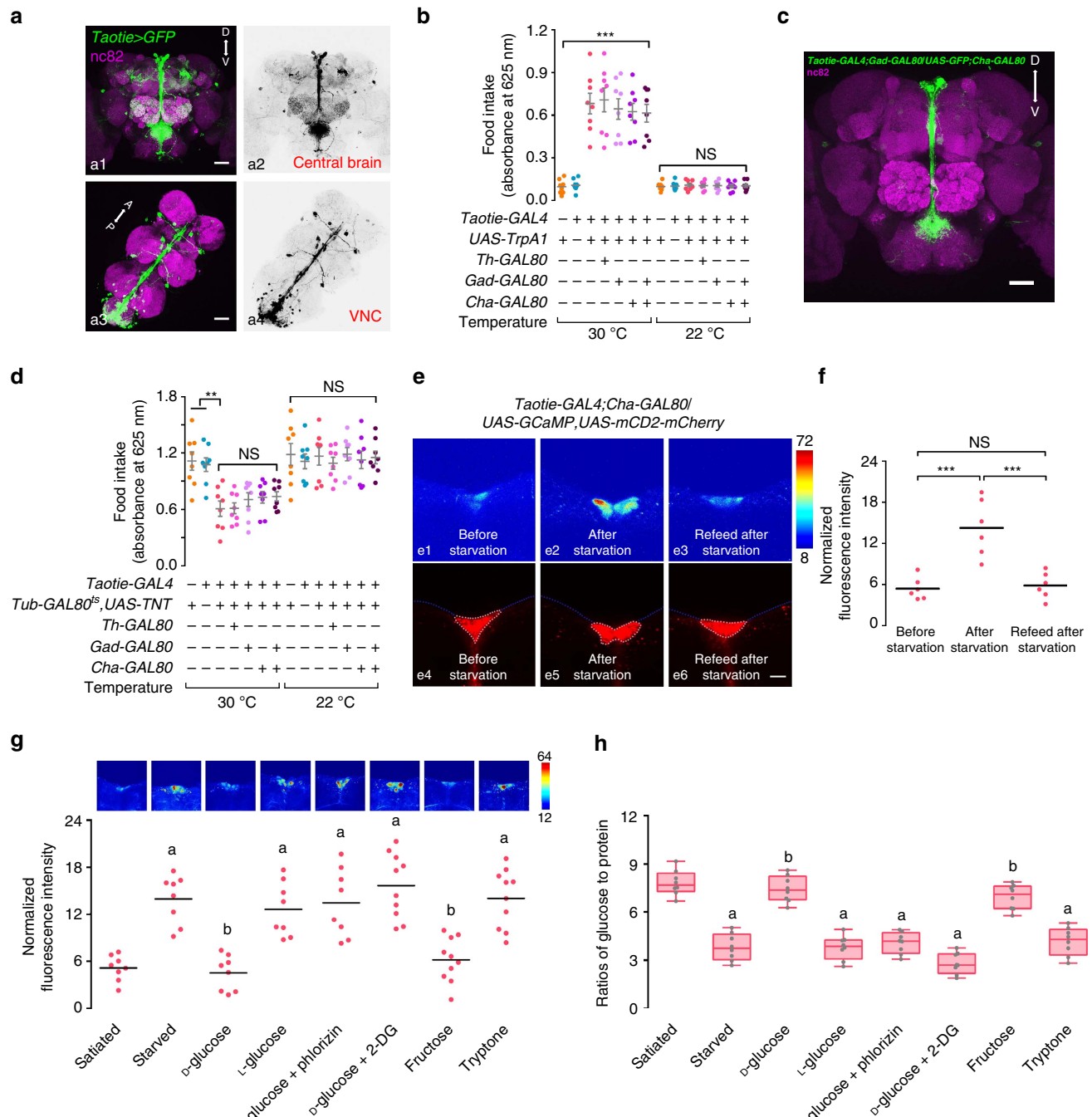

**Figure 6 | Activity of Taotie neurons in the central brain is responsible for regulating feeding behaviours according to hunger/satiety states. (a)** Brain and VNC of *Taotie-GAL4/UAS-mCD8-GFP* (green) flies immunostained with the neuropil marker nc82 (magenta, as counter stain) (a1 and a3); a2 and a4 show GFP signals only. Scale bar, 50 μm. **(b)** Flies with *Taotie-GAL4, UAS-TrpA1,* and different GAL80s were tested for food intake under satiated conditions ($N = 8$). **(c)** Expression pattern of *Taotie-GAL4;Gad-GAL80/UAS-mCD8-GFP;Cha-GAL80* (green) counter-stained with nc82 (magenta) in the whole brain. Scale bar, 50 μm. **(d)** Starved flies with *Taotie-GAL4, Tub-GAL80^{ts}, UAS-TNT* and different GAL80s were tested for food-intake ($N = 8$). **(e)** Confocal sections of GCaMP signals in the PI region: before-starvation group (e1, satiated), after-starvation group (e2, starved) and refeed-after-starvation group (e3, satiated). Panels e4–e6 show the corresponding cell clusters in this region, which were identified according to mCherry signal (red). Scale bar, 15 μm. **(f)** Comparison of GCaMP signals normalized to mCherry signals in before-starvation, after-starvation and refeed-after-starvation groups ($N = 6$). Black bars indicate average signal intensity of each group. **(g)** Comparison of normalized GCaMP signals in *Taotie-GAL4/UAS-GCaMP6s;UAS-RFP* flies treated with different diets ($N = 8$–10). Scale bar, 20 μm. **(h)** Hemolymph glucose levels, normalized to protein levels, in *Taotie-GAL4/UAS-GCaMP6s;UAS-RFP* flies fed with different diets ($N = 8$). All genotypes, temperatures and experimental conditions were as indicated with the plots. In a box and whisker plot, whiskers mark minimum and maximum, box includes 25th – 75th percentile, and the line in box indicates median of the data set. NS indicates not significant ($P > 0.05$); **$P < 0.01$, ***$P < 0.001$ (Student's *t*-test for two group-only comparisons, ANOVA with Bonferroni *post hoc* test for multiple comparisons). For **(g,h)** different letters indicate groups with statistically significant differences to the satiated group. (**a**): $P > 0.05$; (**b**)$P < 0.001$. Error bars indicate s.e.m.

yet palatable) for 24 h, the calcium signals in Taotie neurons were dramatically increased in the L-glucose group, but not in the D-glucose group, suggesting decreased hemolymph glucose level activates Taotie neurons (Fig. 6g,h). Next, we treated flies with drugs to induce hypoglycemia and measured neuronal activities of Taotie neurons (Fig. 6g,h). One drug, phlorizin, is a blocker of glucose transporter to inhibit the entry of glucose into the hemolymph. Another drug is 2-Deoxy-D-glucose (2-DG), which is an inhibitor of glucose metabolism. As expected, decreasing blood glucose levels by both drugs elevated the activity of Taotie neurons (Fig. 6g,h). Hemolymph fructose levels are thought to serve as an indicator for the consumption of nutritious carbohydrate[24]. Similar to D-glucose, feeding flies with fructose did not change the activity of Taotie neurons (Fig. 6g,h). Moreover, feeding flies with protein-rich food evoked high neuronal activity in Taotie neurons (Fig. 6g,h). These results strongly suggest that neural activity of Taotie neurons are regulated by hemolymph glycemia levels. To confirm our hypothesis, we performed re-feeding experiments, whereby flies were fed with different sugars for 1 h after 24 h of starvation. As shown in Supplementary Fig. 14b,c, caloric sugars effectively suppressed neuronal activity of Taotie neurons, but non-caloric sugars and proteins-rich food were not. Thus, our data demonstrate that the level of hemolymph glycemia is a crucial physiological factor to regulate activity of Taotie neurons, further supporting the idea that Taotie neurons serve as an important mediator, both to receive physiological signals and coordinately command feeding behaviour.

**Taotie neurons control insulin secretion**. Next we investigated the mechanisms for the feeding centre presented by Taotie neurons to gate feeding behaviour and control energy homeostasis. Our behaviour-based appetite screen did not reveal possible players for the evoked feeding behaviour although we included most of neurotransmitters and neuropeptide systems in the screen. In addition, Taotie-GAL4 neurons did not overlap with neurons expressing NPF, an orthologue of the mammalian neuropeptide Y[43] (Supplementary Fig. 15d). We noticed that the predominant cluster of Taotie neurons (10–12 neurons) was in the PI region, which is the main neurosecretory area of Drosophila brain (Fig. 7a). The PI region contains 10–14 insulin producing cells (IPCs), secreting Drosophila insulin-like peptides (Dilps, the homologue of human insulin). These IPCs secrete three (Dilp2, Dilp3, and Dilp5) of eight known Dilp isoforms[44]. The Taotie-positive PI neurons were completely complementary to the IPCs in the same region (Fig. 7a and Supplementary Fig. 15a–c and Supplementary Movie 1). A previous study indicated that in animals placed under nutrient deprivation, Dilps secretion from the IPCs is inhibited, thus resulting in accumulation of Dilps in these cells[45]. Considering that the activities of Taotie neurons represent neurological hunger state and Taotie neurons juxtapose with IPC cells, we investigated the relationship between activity of Taotie neurons and insulin signalling in satiated and starved flies. As shown in Fig. 7b, starvation induced high levels of insulin accumulating in the IPCs cells[45]. Subsequently, feeding of hungry flies reduced Dilps accumulation to levels similar to those in satiated flies, demonstrating that the level of insulin in IPCs is negatively correlated with satiety states of animals (Fig. 7b). Accordingly, activation of insulin-expressing neurons resulted in reduced food-intake, suggesting that insulin functions as a satiety signal that suppresses feeding (Supplementary Fig. 16a). Inactivation of IPCs alone is not sufficient to evoke over-feeding, consistent with our hypothesis of insulin as a satiety command primarily controlling the satiation states of an animal (Supplementary Fig. 16b,c). Intriguingly, overexpression of constitutively active and dominant

negative insulin receptors with Taotie-GAL4 showed no significant effects on feeding in both starved and satiated flies, suggesting that the Taotie neurons are not directly under the control of the insulin pathway to gate feeding behaviour (Supplementary Fig. 16d).

We next asked whether Taotie neurons function as a 'regulator' to control insulin secretion in corresponding to physiological states. Surprisingly, activation of Taotie neurons in satiated flies resulted in accumulation of insulin signals in the IPC cells, mimicking the conditions of starvation (Fig. 7c). Therefore, in addition to activating hungry circuits, Taotie neurons also act upstream of the insulin signal to suppress satiation circuits. Inactivation of Taotie neurons reduced the accumulation of insulin in starved flies, further demonstrating a negative regulation of Taotie neurons on insulin secretion in both starved and satiated states (Fig. 7d). These results suggest that modulating the activity of Taotie neurons forces changes in the neuroendocrine status, even if these changes are in contradiction to the true/actual physiological state of the animal. It was previously shown that insulin signals regulate energy balance in adult flies[35]. We therefore evaluated the effect of chronic activation of Taotie neurons on insulin signalling while the animals were provided with sufficient food. After 4 days of persistent activation, the accumulation of insulin in Taotie > CsChrimson flies was much higher than controls, despite the fact that the animals had been ingesting higher amounts of food than normal (Fig. 2b and Supplementary Fig. 17). Furthermore, the insulin signals reverted back to normal levels within 4 days after withdrawing the source of activation (Supplementary Fig. 17). These results demonstrate that Taotie neurons act upstream of insulin system to control insulin secretion and regulate both feeding behaviour and energy balance.

It has been shown that in the PI region, another population of neurosecretory cells, the Diuretic hormone 44 (Dh44) producing neurons, are specifically activated by nutritive sugars in starved flies to modulate food choice[25]. We conducted a series of experiments to distinguish Dh44 neurons and Taotie neurons. First, double-labelling experiments showed that only 1–2 Taotie-positive neurons were co-localized with the Dh44 positive cells in the PI region (Supplementary Fig. 18a,b and Supplementary Movie 2). Next, we specifically manipulated the Taotie-positive PI neurons in the PI region. In order to prevent the Taotie positive and Dh44 positive neurons from activation by Taotie-GAL4, we expressed GAL80 in Dh44 neurons, which effectively suppresses the GAL4 activities in the same cells. This approach allowed us to visualize and manipulate those neurons that were labelled by Taotie-GAL4 alone (Supplementary Fig. 18c,d). Behavioural tests with the remaining Taotie neurons in the PI region indicated that these neurons are both necessary and sufficient for controlling feeding behaviour (Supplementary Fig. 18e,f). Furthermore, as activation of Dh44 neurons did not result in changes of feeding behaviour, we next conducted neural epistasis to examine the behavioural consequence after co-activation of Taotie neurons and Dh44 neurons (Supplementary Fig. 18g). Interestingly, activation of Dh44 neurons did not interfere with the overfeeding behaviour evoked by Taotie neurons. This result is distinct from our epistatic interactions with co-activation of Taotie neurons and other satiated signals in flies (AstA, Dilp2 and Gr43a)[14,24,46] (Supplementary Fig. 19). Together, these results further support the idea that Taotie neurons mediate transition between hunger and satiety states through interfacing neurological, physiological and neuroendocrine signals (Fig. 7e).

**Discussion**
In this study, we identified a population of neurons whose neural activity is closely associated with the physiological states of

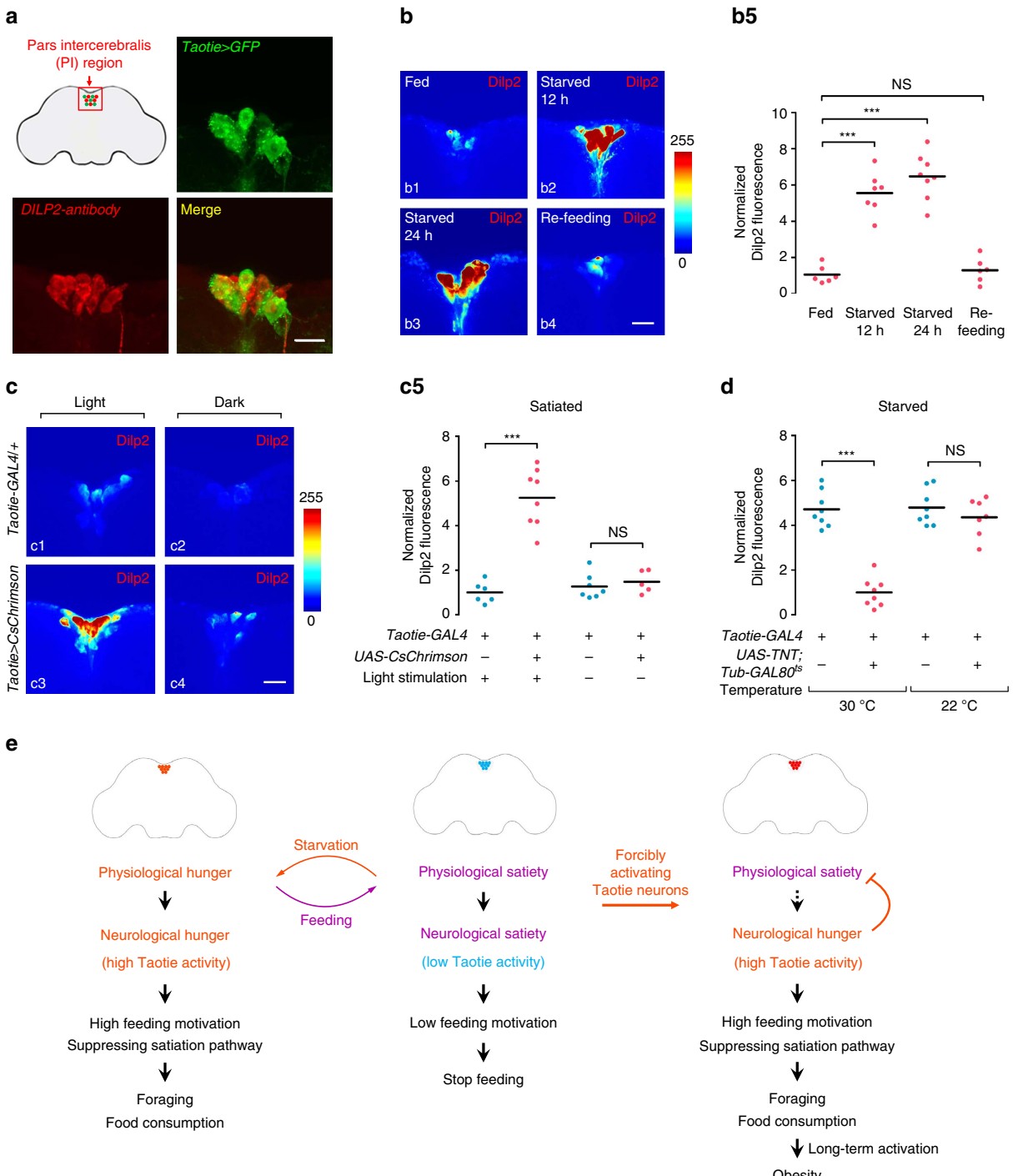

**Figure 7 | Taotie neurons regulate insulin secretion in satiated and starved flies. (a)** A diagram of neurosecretory cells located in the PI region, showing that Taotie neurons (green) in PI region are complementary to the insulin-producing cells (labeled by anti-Dilp2 antibodies, red) in *Taotie-GAL4/UAS-mCD8-GFP* flies. Scale bar, 10 μm. **(b)** Normalized fluorescence intensities of Dilp2 by anti-Dilp2 antibodies in fed, starved, and re-feeding flies (b1–b4), showing that starvation induces insulin accumulation in IPCs. Scale bar, 20 μm. The fluorescence intensities of Dilp2 in b5 were normalized to the fed group (N = 6–8). **(c)** In satiated flies, insulin accumulation occurs only when Taotie neurons are activated. Signals in *Taotie-GAL4/UAS-CsChrimson* and genetic control flies were measured after 1h light stimulation (c1 and c3). Experimental control flies were kept in darkness (c2 and c4). Scale bar, 20 μm. The fluorescence intensities of Dilp2 in c5 were normalized to that of *Taotie-GAL4/ +* flies with light stimulation (c1) (N = 5–8). **(d)** In starved flies, silencing Taotie neurons prevents insulin accumulation. Flies were subjected to starvation in 30 °C for 24 h to inactivate Taotie neurons. The fluorescence intensities of Dilp2 were normalized to that of *Taotie-GAL4/UAS-TNT;Tub-GAL80^{ts}* flies at 30 °C (N = 7–8). **(e)** Working model of Taotie neurons: Taotie neurons in the PI region are normally quiescent or suppressed by internal physiological states when flies are satiated (middle). However, as flies become hungry (left), these neurons are activated by physiological signals to enter into a 'neurological hunger' state, which in turn elevates feeding motivation to activate downstream feeding sub-programmes. Forcibly activating these neurons promotes neurological hunger and suppress satiation in physiological satiated flies, thus results in elevated appetite, hyperphagia and obesity (right). All genotypes, temperatures and experimental conditions are indicated with the plots. NS indicates not significant ($P > 0.05$); ***$P < 0.001$ (Student's *t*-test for two group-only comparisons, ANOVA with Bonferroni *post hoc* test for multiple comparisons).

hunger and satiety. Manipulating the activity of these neurons directly elicited hunger- or satiation-related feeding behaviour, regardless of the physiological need of the flies. Chronic activation of these neurons resulted in flies exhibiting obese phenotypes with characteristics similar to human. Our results revealed the presence of a prescriptive feeding centre in the *Drosophila* brain, thus shedding light on the relationship of appetite, hunger/satiety, and obesity.

Besides external factors such as food composition and palatability, the conscious or subconscious desire for excess amounts of food, regardless of actual physiological status, is becoming increasingly responsible for obesity-inducing hyperphagia[4]. In *Drosophila*, details of the intricate relationship between brain and body for maintaining a healthy energy homeostasis have just begun to emerge[25,35]. Previous studies showed that the communication between brain and fat tissue is important for energy homeostasis, and that manipulation of neural activity in the brain alters the body fat storage[47]. However, the precise nature of the neural substrate regulating the energy balance has long been awaiting elucidation. Association of *Taotie-GAL4* neurons with obese phenotypes in this study provides a much-needed genetic handle to understand the key neural circuits controlling feeding behaviour. Furthermore, our results suggest that altering the activity of certain critical neurons associated with basic physiological states overrides the inhibitory effects of satiation signals on feeding, induces hunger-like appetite, and unbalances the energy homeostasis of the organism, similar to those observed in patients with obesity and obesity-related diseases[29].

Human obesity is a chronic disease for which few effective pharmacological treatments exist[5]. The prevalence of quick weight regain, even after successful weigh-reduction treatment, adds to the severity of obesity in humans[48]. Interestingly, our results suggest that, despite the intact repressive mechanisms of physiological satiety, long-term activation of hyperphagia-inducing neurons in the brain results in obesity, suggesting that a targeted approach to treating the causes of obesity is important. Moreover, we showed that modulating activity level of Taotie neurons fine-tunes the body weight. Furthermore, reversal from an obese to non-obese phenotype following restoration of normal neuronal activity in the brain suggests that, once the activity of obese-responsible neurons is established again at normal levels, the body actively rectifies the energy imbalance to its previously normal physiological status. Therefore, activation of Taotie neurons will serve as an important *Drosophila* model for the study of the processes related to physiological recovery following episodes of obesity.

Similar to mammals, fruit flies utilize multiple conserved molecular signals to regulate the energy balance, especially under conditions of varying availability of food[35,49–51]. Interruptions on such signal pathways result in unbalanced energy homeostasis that would lead to obesity phenotypes. For instance, ablation of IPCs generates phenotypes similar to type 1 diabetes[50], while flies fed with high-sucrose or high fat diet exhibit hallmarks of type 2 diabetes, including hypertriglyceridemia, hyperglycemia, insulin resistance and obesity[51,52]. Our data show that activities of Taotie neurons regulate the release of insulin peptides. Energy homeostasis was further skewed in flies with activated Taotie neurons on high energy diet, while suppressing Taotie neurons reduces the high level of glucose and triglyceride induced by high sugar diet to normal, suggesting an intriguing role of Taotie neurons on controlling molecular events related to energy balance. Diabetes and insulin resistance typically associate with chronic low-grade inflammation[32]. Interestingly, the obese flies induced by Taotie neuron activation also display impaired insulin-related signals and increased expression of cytokines,

further suggesting conserved mechanisms of obesity-related metabolic disorders across species.

A series of neuroendocrinological events and behavioural changes contribute to a normal hunger response[53]. At its onset, hunger causes a wide range of changes in neuronal activity throughout the brain, with multiple neural circuits being linked to or implicated in starvation (or hunger) in *Drosophila*. In the periphery, both olfactory and gustatory inputs related to feeding are affected by hunger[2,54,55]. In the brain, multiple neural transmitters (Serotonin, Dopamine and Octopamine) and neuropeptides (dNPF, sNPF, Drosulfakinin, Tachykinin, CCHamide-2 and Allatostatin A) participate in different, sometimes opposite, aspects of feeding response upon starvation[13,14,16,17,21,39,43,46,56]. Thus, to improve our understanding of the mechanisms of hunger, it is important to distinguish whether a neural circuit associated with hunger-induced behaviour is the cause or the consequence of hunger sensation. The behaviour-based screens taken by previous studies as well as this study strongly demonstrate the feasibility of applying gain-of-function approach to obtain a genetic handle crucial for the initial phase of investigating hunger mechanisms in *Drosophila*[13,21,47]. Our appetite screen covers many neurons and brain regions as well as biogenic amines and neuropeptidergic systems. We found that only the activation of *Taotie-GAL4* neurons evokes ravenous feeding behaviour, and increases feeding motivation in satiated flies (Supplementary Fig. 20).

Within Taotie neurons, we identified *amon*, a proprotein convertase important for producing neuropeptides[42]. *Amon* knockdown in Taotie neurons suppresses overeating behaviour evoked by Taotie neurons activation, suggesting neuropeptides are required for the proper functions of Taotie neurons on feeding regulation. In *Drosophila*, there are at least 40 neuropeptides and more corresponding receptors[20]. The identity of the causative neuropeptides in Taotie neurons, although currently unknown, would help to further investigate their conserved functions in feeding controls. Analysis of the hierarchical relationship of Taotie neurons with other feeding control neurons indicates that Taotie neurons interact antagonistically with those promoting satiety to elicit cohesive feeding behaviour. Further exploration of the hierarchy of feeding-related neural circuits should contribute to understand how feeding behaviour is coordinately and collectively regulated by neurological, physiological and neuroendocrine signals in animals.

Control of feeding for maintaining a healthy energy home-ostasis involves intertwining neural and physiological processes; as a consequence, hunger consists of both physiological and neurological changes in the brain and body[29]. While changes in neural activity are fast, physiological changes of the body occur at a much lower speed. Due to the difficulties of separating physiological hunger from neurological hunger, it was not possible to discern whether neurological hunger is persisting (as a condition or state) or transient (as an event, the instantaneous result from processing physiological signals). We showed that elevated appetite is a consequence of evoking an endogenous neural response toward hunger. Interestingly, the temporal separation of activation of Taotie neurons and the subsequent feeding action argues that the activation, although only transient, evokes a persistent hunger state, which then initiates motor programmes to feed under suitable external contexts, very similar to other persistent behavioural states such as aggression or sexual arousal[57,58]. Our neuroimaging data further suggest that the endogenous hunger state of the animals is represented by the activities of Taotie neurons in the neuroendocrine region of the brain. Taken together, the function

of these neurons is analogous to that of a general thermostat. A thermostat measures temperature, and also compares and adjusts temperature to a set point by turning on/off the heating apparatus. In comparison, the neurons investigated here monitor the physiological status of the body, and by doing so, they make decisions about either commencing or terminating feeding.

In mammals, the hypothalamic brain centres receive information of the physiological status through pathways governed by various hormones such as leptin, insulin and ghrelin[59,60]. These centres also command behavioural changes to balance feeding and energy metabolism for maintaining a constant body weight[26,60]. Interestingly, optogenetic stimulation of agouti-related peptide AGRP neurons in the hypothalamus is sufficient to evoke voracious feeding in satiated mice, while ablation of AGRP neurons induces anorexia in starved mice[26,61]. Long-term activation of AGRP neurons results in elevated appetite and obesity[62]. Furthermore, in mice, the neural activity of AGRP neurons dramatically increases during fasting[63]. Intriguingly, recent study found that transient activation of AGRP neurons evokes a persistent hunger state[64]. The Taotie neurons in flies exhibit a striking similarity to those starvation-sensitive AGRP neurons in mice, suggesting that the principles controlling feeding behaviour are conserved across the animal kingdom.

As feeding behaviour is crucial for animal survival, a brain must utilize multiple and possibly redundant neural circuits to coordinate a hunger response. For example, in *Drosophila*, starvation enhances behavioural sensitivity to sugar partly via increased dopamine levels, and the activity of dopaminergic neurons is altered by the hungry state[16,17]. Another neurotransmitter, octopamine, is crucial for starvation-induced hyperactivity, but not required for the regulation of energy intake during starvation[56]. Recent studies indicated that a subset of serotonergic neurons, the R4 neurons in ellipsoid body and four neurons in SEZ, participate in hunger sensation in *Drosophila*[21,65,66]. How much these circuits contribute to various aspects of hunger sensation, especially how they interact dynamically and collectively to control feeding, remains to be revealed.

The unique yet essential features of Taotie neurons render them good candidates as part of a prescriptive feeding centre that represents the hunger/satiety states in the *Drosophila* brain. Interestingly, the key *Taotie-GAL4* neurons form a cluster in the PI region. Our imaging data demonstrate that hypoglycemia serves as an important physiological factor to activate these neurons to confer the hunger status in the brain. Thus, the activity of these neurons both represents physiological status and initiates a series of behavioural changes during hunger, essentially functioning as an interface for physiological status and neurological response. The PI region is known to regulate many physiological events and states including sleep, circadian rhythms, social behaviour, nutrient sensing, lipid and carbohydrate metabolism[25,35,50,67]. Thus, our results provide overwhelming evidence that the PI region plays an essential role for feeding behaviour and physiology balance in *Drosophila*, which is probably similar to the centres of energy homeostasis in mammals.

Inside this proposed feeding centre, Taotie neurons show a complementary expression pattern with the neighbouring IPCs, and Taotie neurons negatively regulate the release of insulin from IPCs. Our results suggest that, as in human, insulin signalling promotes satiety in *Drosophila*. In support of our hypothesis, DSK (Cholecystokinin-like peptide) in the IPCs was suggested earlier to mediate a satiety signal in *Drosophila*[46]. In addition, the functional leptin homologue Unpaired 2 (Upd2) regulates systemic growth and energy balance in *Drosophila*[35]. Upd2 released from the fat body conveyed nutrient status to IPCs via a population of GABAergic neurons near the IPCs[35]. Together, these data suggest that insulin release is under multiple upstream regulators. Besides insulin, several receptors of neurotransmitters and neuropeptides are also expressed in the PI region, and together they may regulate feeding and energy homeostasis in a coordinated fashion[67]. Our results suggest that Taotie neurons, located adjacent to IPCs, regulate feeding behaviour and energy homeostasis through controlling insulin secretion in both satiated and starved flies (Fig. 7e). To dissect hunger-driven behaviour at both the molecular and cellular level, it will be crucial to divide the PI region into different sub-regions, according to their specific functions in the regulation of physiological homeostasis.

## Methods

**Fly stocks.** Flies were reared on standard media at 22 °C or 25 °C, 60% relative humidity, under a 12 h light:12 h dark regime unless otherwise indicated. *Canton S* was used as wild-type control. The GAL4 collection used for the behaviour screen originated from our lab's previous collection. The *Taotie-GAL4* line was provided by J. Carlson, and was generated by promoter fusion, containing a region approximately 4.7 kb upstream of the translation initiation point of the *Gr28b.b* gene. *UAS-TrpA1* (temperature sensitive) (BL26263 and BL26264), *Tub-GAL80ts* (temperature sensitive) (BL7017 and BL7019), *UAS-InR-ACT* (constitutively active), *UAS-InR-DN* (dominant negative), *AstA-GAL4* (BL51979), *Dh44-LexA* (BL54089), *LexAop-GAL80* (BL32213), *UAS-GCaMP6s* (BL42746), *UAS-Amon^RNAi* (BL29010), *UAS-Amon* (BL29008), *UAS-ReaChR* (BL53749), *UAS-CsChrimson* (BL55135 and BL55136), *UAS-DenMark* (BL33062) and *Gr28b* mutant (BL24190) were obtained from the Bloomington stock centre. Additionally, *Dh44-GAL4* (gift from G. Suh), *Gr43a^GAL4* (gift from H. Amrein), *Gr5a-GAL4* (gift from K. Scott), *Gr64e-GAL4* and *Gr64f-GAL4* (gifts from J. Carlson), *Dilp2-mCherry* (gift from A. Sehgal), *UAS-GCaMP3.0,UAS-mCD2-mCherry* (gift from Y.-N. Jan), *UAS-Gr28b.b* (gift from P. Garrity), *Dilp2-GAL4, UAS-Shibire^ts, UAS-syt-GFP, LexAop-mCD8-RFP* (gifts from Y. Rao), *Elav-GAL80* and *Tsh-GAL80* (gifts from L. Vosshall), *UAS-TNT;Tub-GAL80ts, Tub-GAL80ts;UAS-Kir2.1, Th-GAL80, Cha-GAL80,* and *Gad-GAL80* were kindly provided by Y. Li and A. Guo.

**Chemicals.** Sucrose (S7903), D-glucose (G7528), L-glucose (G5500), D-arabinose (10850), denatonium benzoate (D5765) and 2-Deoxy-D-glucose (D-6134) were purchased from Sigma-Aldrich. Sucralose (65684B) and phlorizin (69592A) were purchased from Adamas. Trehalose (K480), sorbitol (0691), D-fructose (0226) and D-ribose (0671) were purchased from AMRESCO. Yeast extract (LP0021) and tryptone (LP0042) were from OXOID. Quinine (Q0030) was purchased from TCI AMERICA.

**Food-intake assay.** Food-intake assays were performed using 5 to 10-days-old adult female flies. All feeding experiments were conducted at the same time of the day, namely around Zeitgeber time 4–6. The bottom of an outside container (a vial with diameter of 31.8 mm and height of 80 mm) was covered with 1% agarose as a water source to keep the inside of the container humidified. At the centre of bottom of the container, we placed a feeding chamber (diameter: 13.64 mm and height: 7.5 mm) containing different food sources. Most of the behavioural tests used sucrose (100 mM) as food source, except for other specific descriptions. A blue food dye (Erioglaucine Disodium Salt, Sigma 861146) was added to the food at 0.5% (w/v). Before behavioural experiments, vials were allowed to reach equilibrium at the permissive temperature (30 °C) for 2 h to ensure the temperature gradient is maintained within ± 1 °C from its floor to ceiling at experimental temperatures. Twenty flies were carefully transferred into one vial to start the feeding process. Feeding was interrupted at different time points by freezing the vials at − 80 °C. Flies were homogenized in 500 μl PBS buffer with 1% Triton X-100 in 1.5 ml Eppendorf tubes, and centrifuged (13,000 r.p.m.) for 30 min to clear the debris. After centrifugation, the absorbance of the supernatant was measured at 630 nm (A630). The background absorbance for supernatants from flies fed with regular food was subtracted from the absorbance of the supernatant from blue food-fed flies. The net absorbance reflected the amount of food ingested. For surgical experiments, the front leg and middle leg tarsi were surgically removed bilaterally under anesthesia 2–4 days prior to behavioural test.

**Quantification of food ingestion of individual flies.** To classify and score the proportion of flies with visually detectable dye in their abdomen, the fraction of flies containing blue dye following a 1-h feeding assay was analysed under a dissecting microscope. Flies lacking visual cues of feeding were given a score of 0, those with only detectable blue dye in their abdomen were given a score of 1, those with less than half of their abdomen volume were given a score of 2, those with more than half blue abdomen were given a score of 3 (also see Fig. 1c). Images were taken with a stereo Leica M205 FA microscope using a Leica Application Suite software.

**Hunger-driven food odour attraction.** To measure the hunger-driven food odour attraction as an indicator for increased feeding motivation, a 16 cm transparent plastic tube was used. The two ends of the tube were connected to different odour sources. All tests were performed around Zeitgeber time 5–8. After a brief anesthesia, 30 adult female flies (5 to 10-days-old) were transferred into a tube; the vapours of apple cider vinegar and water (as control) were provided to either end of the tube. The fly distribution was captured by a camera that took a photo every 5 s. Photos from 5–10 min were used to perform the analysis. To analyse the distribution of flies from the images, a MATLAB programme was written to remove background and calculate the position of flies.

Preference index was calculated according to the following equation: preference index $= (N^+ - N^-)/(N^+ + N^-)$, where $N^+$ = numbers of flies located on the side with food odour, and $N^-$ = numbers of flies located on the side with water. A preference index of 0 indicates no difference between two sides, whereas a preference index of 1 indicates that all flies are on the side with the food odour.

**PER assay.** For detection by PER assay, flies were fed on regular food, or starved in a vial with 1% agarose at 30 °C or 22 °C. Flies were anaesthetized by chilling in a tube standing on ice, and gently mounted onto yellow pipette tips. For all experiments, immobilized flies were allowed to drink water until they ceased extending their proboscis and then were tested for their response to the stimulation of sucrose solution. Sucrose solutions were presented to each fly once, in order of increasing concentration. The labellum was briefly touched with a paper strip soaked with sucrose solution, so that flies could not drink the solution during the entire test. Only full extensions of proboscis, but not partial extension, were scored as positive result. For *UAS-TrpA1* experiments, flies were warmed up to 30 °C and incubated for 10 min, and behavioural test were then performed at 30 °C. For observation of spontaneous proboscis extensions behaviour, single flies were transferred to a pre-heated box at 30 °C, and times of proboscis extensions were calculated within 30 s. For each experimental condition, we used 5–15 flies to repeat independently at least three times. All PER experiments were performed around Zeitgeber time 4–6.

**Two-way choice assay.** To measure the ability of gustatory discrimination by a two-way choice assay, 50 female flies were induced to express TNT for 48 h at 30 °C and then starved for 20 h on 1% agarose at 30 °C before subjecting to the behavioural test. Flies were then placed into 72-well microtitre dishes with two types of test mixtures filled in alternating wells. The test mixture contained either blue dye (0.125 mg/ml Brilliant Blue FCF) or red dye (0.2 mg/ml sulforhodamine B, Sigma) in 1% agarose with different concentrations of sucrose to be tested. Flies were allowed to feed for 90 min at room temperature in the dark before they were frozen. The flies with blue ($N^B$), red ($N^R$) or purple ($N^P$) abdomens were assessed by visual inspection. Preference Index, PI, was calculated according to the following equation: $PI = (N^B + 0.5N^P)/N^{Total}$ or $(N^R + 0.5N^P)/N^{Total}$. Thus, a PI of 0.5 indicates no preference whereas a PI of 0.5–1 indicates a preference. The blue and red dyes did not cause preference bias, as no differences in PIs were detected by switching the dyes.

**Inducible inactivation.** The tetanus toxin light chain TNT (or the inward-rectifying potassium channel Kir2.1 (ref. 68)) was expressed exclusively in the adult stage using the TARGET system to avoid developmental effects. Flies with *Taotie-GAL4, UAS-TNT* (or *UAS-Kir2.1*) and *Tub-GAL80^{ts}* were reared at 22 °C, and collected within 4–8 days after eclosion. Two days temperature shift to 30 °C inhibited GAL80 activity in order to induce TNT (or Kir2.1) expression. Flies were starved at 30 °C for another 24 h with 1% agarose and then returned to 22 °C for 2 h prior to testing. Control (non-induced) flies remained at 22 °C. Behavioural test was performed at room temperature. Amount of food intake within 24 h of starvation at 30 °C was similar to that at 36–40 h of starvation at 22 °C. Thus, for all experiments using the TARGET system, induced groups starved at 30 °C for 24 h and non-induced control groups starved at 22 °C for 36–40 h, to obtain similar metabolic rates at different temperatures.

**Optogenetic stimulation.** Newly enclosed flies were collected and transferred into a vial with regular food containing 400 μM all-trans-retinal (Sigma R2500). Vials were wrapped in aluminum foil to protect the retinal from light, and were then kept in a 25 °C, 60% humidity incubator for 4–6 days. All experiments were performed under a dim light to avoid the effect caused by ambient light. During behavioural tests, an array of orange LED (emission maximum: 625 nm) was placed under the vials as the source of stimulation. The maximum intensities of LEDs in this setup caused *Taotie-GAL4/UAS-CsChrimson* flies to paralyse with cessation of locomotion and loss of postural control. We used an intermediate intensity to stimulate *Taotie-GAL4/UAS-CsChrimson* flies during the feeding test. The paralysis phenotype did not occur when using *Cha-GAL80* and *Gad-GAL80* to restrict the expression of *Taotie-GAL4*.

In laser targeting activation experiments, we immobilized *Taotie>CsChrimson* flies on glass slides, and used a blue laser (470 nm) to active CsChrimson. A foil plate was used to restrict the exposure area to laser light, focusing only on head or thorax without illuminating other regions. During the experiment, we first stimulated either the brain or the thorax for 10 min with the laser, then delivered

sucrose solutions to measure the PER response. Light intensity was measured by a Spectrometer (CCS200/M, THORLABS).

**Metabolic measurement.** For TAG (Triglyceride) assays, 10 adult female flies were decapitated and homogenized in 300 μl PBS with 0.5% Tween-20, from which 30 μl was transferred for measuring protein levels. The remaining liquid was immediately incubated at 70 °C for 5 min. Heat-treated homogenates (40 μl) were incubated with either 40 μl PBS or Triglyceride Reagent (Sigma T2449) for 30 min at 37 °C, after which the samples were centrifuged at maximum speed for 5 min. 30 μl of the supernatants was transferred into a 96-well plate and incubated with 100 μl of Free Glycerol Reagent (Sigma F6428) for 5 min at 37 °C. Samples were assayed using a microplate spectrophotometer at 540 nm. TAG amounts were determined from the total glycerol present in the sample treated with Triglyceride reagent. TAG levels were normalized to protein amounts in each homogenate using a Bradford assay (Bio-Rad 500-0205 AND 500-0206).

The concentrations of total circulating glucose were measured from hemolymph. Ten female flies were decapitated, and their hemolymph was extracted by centrifugation (2 min, 14,000 r.p.m. at 4 °C). A volume of 0.5 μl of hemolymph was mixed with 100 μl of the Glucose Assay Kit (Sigma: GAHK20) adjusted to pH 6.8. The pig kidney trehalase (Sigma #T8778) (1:500) was diluted into the mixture and incubated at 37 °C for 16 h. The ratios of glucose to protein were normalized using a Bradford assay (Bio-Rad 500-0205 AND 500-0206).

**Histology.** For cryosectioning, *Taotie-GAL4/UAS-CsChrimson* female flies aged 4–6 days were fed with all-trans-retinal (400 μM). After removal of wings and legs, flies were embedded in OCT medium prior to freezing at − 40 °C. 60 μm sections were cut at − 14 °C using a Thermo Cryotome E, and were then transferred onto coated slides. For lipid staining, sections were dried for 5 min at 30 °C prior to staining. Cryosections were fixed for 20 min in 4% paraformaldehyde solution. Specimens were washed three times in 1xPBS and stained with BODIPY (Life Technologies #D3922) for 40 min at room temperature. The specimens were washed again three times with 1xPBS, and mounted using glycerol.

**Calcium imaging.** Previously established methods for calcium imaging was used with minor modifications[24,25]. In brief, flies were divided into two groups and starved or fed with different diets for 24 h on 1% agarose, one group was directly used for calcium imaging and the other group was allowed to feed *ad libitum* on different diets for 1 h before the preparation for imaging. Flies were randomly picked from their housing vials for all experiments, and experiments were performed at Zeitgeber time 4–6. For imaging experiments, flies were first briefly anesthetized on ice, before the brains were quickly dissected at room temperature into a sugar-free ringer solution (5 mM HEPES, pH 7.2, 130 mM Nacl, 5 mM KCl, 2 mM CaCl₂, 2 mM MgCl₂). An O-ring was glued to a glass slide, containing a 5 mm diameter area in the middle to allow a small reservoir to form. Brains were placed into the reservoir and covered with 20 μl ringer solution. GCaMP fluorescence was measured following excitation of the sample with a 488-nm laser. Images were acquired at $512 \times 512$ pixels. Focal planes were selected for regions of interest, based on mCherry or RFP signal. The imaging rate was 0.11 frames/s. To reduce background noise, three frames were averaged. To analyse the resulting calcium imaging data, a MATLAB programme was written. This programme used a modified Otsu's method to choose a threshold of contrast by minimizing the intraclass variance of the foreground and background pixels in the red fluorescence channel (mCherry signals). The threshold was used to automatically identify the areas occupied by cell bodies as the region of interest (ROI). The programme further calculated green fluorescence intensity (GCaMP signals) from the ROI regions. Finally, the ratios of green fluorescence intensity and red fluorescence intensity were calculated and compared between the before-starvation group, starvation group and refeed-after-starvation group. All sugars in the diets for calcium imaging experiment were used at 200 mM, except sucrose at 100 mM. For drugs of experiment, 2-DG (Sigma) was used at 400 mM and Phlorizin (Adamas) at 10 mM. All calcium imaging experiments were performed with 10–14-days-old female *Taotie-GAL4;Cha-GAL80/UAS-GCaMP3.0,UAS-mCD2-mCherry* flies and *Taotie-GAL4/UAS-GCaMP6s;UAS-RFP* flies.

**Immunohistochemistry.** Dissection of intact brains of adult female flies was performed in cold PBS and fixed in 4% fresh paraformaldehyde solution for 3–4 h on ice. The tissues were then washed with PBT (0.1% Triton X-100 in 1xPBS) five times (15 min each), blocked for 30 min with PBT containing 5% normal goat serum, and incubated with a primary antibody in a blocking buffer for 24 h at 4 °C. After washing with PBT five times, the tissues were incubated with a secondary antibody in PBT for 24 h at 4 °C. All of the fluorescent images were collected using a Leica confocal microscope SP8 and processed with ImageJ, Adobe Photoshop and MATLAB. Antibodies used included mouse nc82 (1:50, DSHB), rabbit anti-NPF (1:2,000, a gift from P. Shen[69]), rats anti-DILP2 (1:400, a gift from P. Léopold[45]).

**Fluorescence quantification of insulin.** Fluorescence intensity of Dilp2 was performed using 4 to 8-days-old and population-density-matched adult female flies. Brains were quickly dissected after each experiment, and stained with

anti-Dilp2 antibodies. Images were captured with a Leica confocal microscope SP8, using identical laser power and scan settings for each experiment. The mean fluorescence intensity was calculated from the total Z stack using ImageJ software.

**RNA purification and real-time quantitative RT-PCR analysis.** Total RNA of whole flies or brains (for *dilp2*, *dilp3* and *dilp5* experiments) were extracted from flies with Trizol reagent (Life Technology), and 1 μg total RNA was reverse transcribed into cDNA using Reverse Transcription Kit (Promega A5001). Quantitative real-time PCR was carried out using EVA green (Biotium 31000) with PCR kit, and data were analysed using a Opticon Monitor 2 software. Samples were collected and analysed in triplicate. Rp49 was used as the normalization reference. Relative differences in gene expression levels were quantified as $2^{\Delta Ct}$ ($\Delta Ct$ is the difference of the Ct values of the gene of interest and Rp49). The qPCR primers used in this study are listed in Supplementary Table 1.

**Statistics.** Statistical analysis was performed using Prism (GraphPad Software). All experiments were performed in parallel with both experimental and control genotypes. For a box and whisker plot, all data points in a data set were plotted, where whiskers mark the minimum and maximum of the data set, box includes data from 25th to 75th percentile, and the line within indicates median. When two groups of normally distributed data were compared, we performed a Student's *t*-test. Fisher's exact test was used to analyse binomial data. ANOVA was used to analyse multiple comparisons in normal distributed data. Following ANOVA analysis, the Bonferroni *post hoc* test was conducted to determine statistical significance. Error bars indicate s.e.m.

**Data availability.** The authors declare that all the data supporting the findings of this study are available within the article (and Supplementary Information Files), or available from the authors upon request.

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

## Acknowledgements

We thank all members of the Y. Zhu lab for stimulating discussions of our manuscript. We thank J. Carlson, A. Sehgal, Y.-N. Jan, L. Vosshall, K. Scott, A. Guo, Y. Li, P. Garrity, P. Shen and L. Pierre for providing flies and antibodies used in this study. We thank Y. Rao for generous supply of various fly strains. We thank T. Juelich for linguistic assistance during the preparation of this manuscript.

We thank Mieyan Huang for providing scientific and administrative support in our lab. We thank Shan Gao for establishing the optogenetic stimulation system and helps in image acquisition processing and analysis. We thank Yi Feng for technical help with the behavioural screen. We thank Tianyu Wang for technical assistant with the data analysis. We also thank Yuxi Zheng for drawing the schematic of the 'appetite screen' experiment. Yinpeng Zhan is indebted to Yuxi Zheng for inspiration and support.

This work was supported in part by National Basic Research Program of China (2012CB825504), National Natural Science Foundation of China (91232720 and 31070925), Chinese Academy of Sciences (CAS) (GJHZ201302 and QYZDY-SSW-SMC015), Bill and Melinda Gates Foundation (OPP1119434), and 100-Talents Program of CAS to Y. Zhu, and also Strategic Priority Research Program of CAS (Grant XDB02040002) to L. Liu.

## Author contributions

Y.P.Z. and Y.Z. conceived the study. Y.P.Z. conducted all the experiments and performed data analysis. Y.Z. and L.L. supervised the project. Y.P.Z., L.L. and Y.Z. wrote the manuscript.

## Additional information

**Competing financial interests:** The authors declare no competing financial interests.

