## [Peer Review File · Nature Communications]

Reviewer #1 (Remarks to the Author)

This manuscript describes the identification of a subset of neurons within the adult brain, called Taotie neurons, which control feeding behavior in *Drosophila*. Specifically, the authors demonstrate that these neurons regulate the hunger/satiety signal such that activation of these neurons causes flies to continuously feed despite being fully fed leading to obese phenotypes. The authors further demonstrate that these neurons are not directly controlled by insulin signaling but do coordinate feeding and physiological homeostasis by regulating insulin secretion.

Overall, this is an extremely interesting area of research aimed at understanding the neural networks that regulate appetite and feeding behavior. It has important implications in our understanding and potential treatment of obesity and obesity associated disorders. While several other studies have previously demonstrated a role for the pars intercerebralis in feeding behavior, Taotie neurons appear to overlap with PI but have distinct functions. Similarly, other studies have shown roles for various neurotransmitters and neuropeptides regulating hunger induced behavior after starvation, but this also appears distinct from the ravenous feeding behavior induced by Taotie neuron activation. Thus, there are several novel aspects to this manuscript. However, in some ways the manuscript is a careful analysis of a group of neurons identified in a GAL4 screen. The authors provide very little data regarding the expression pattern of Taotie-GAL4 and no information regarding what signaling pathways are active/required in these neurons, which is the key question.

Appropriate use of statistics and treatment of uncertainties: Yes

Suggested improvements:

- The authors examine the effect of activating Taotie neurons on food preferences. Given the links to obesity and diabetes, it would have been useful to determine how these flies respond to high sugar and high fat diets. Are these flies more sensitive to these manipulations?
- The authors demonstrate that the obese phenotype is reversible but only examine one timepoint. This analysis should be expanded.
- The authors examine the relationship of Taotie activation with insulin signaling. Is there any link to the fly equivalent of leptin/leptin R (e.g. Upd1/Dome)?
- The authors should consider using RNAi approaches to examine the role of known signaling pathways involved in feeding/appetite regulation using their Taotie-GAL4 driver.

References: appropriate credit to previous work? Yes

Clarity and context: For the most part, the manuscript is clearly written including the introduction and conclusions. However, there are numerous grammatical and spelling errors throughout including the title.

Reviewer #2 (Remarks to the Author)

To optimise survival animals must match their energy intake to their current needs at any point in time. To do this, the brain must regulate feeding behaviour according to the animal's metabolic state. We have some insight into neurons performing some components of this function in mammals and in other organisms, but a circuit level understanding is missing. This study provides an important addition to our understanding of the neuronal control of energy homeostasis and feeding control in *Drosophila*, one of the organisms in which such an understanding could be achieved within a reasonable timeframe.

The authors set out to identify novel neuronal components controlling food intake in *Drosophila*. To do this, they conducted an activation screen to find neurons that increase food intake when activated, and identify one line, Taotie-Gal4, that drives food intake. The authors use bidirectional manipulations of neuronal activity to show that these neurons are both necessary and sufficient for the feeding response to starvation of up to 12 hours. They perform a thorough characterisation of the feeding phenotypes evoked by this line, showing that these neurons control various feeding-related behaviours in a similar way to starvation. Importantly, the authors show that these neurons do not seem to be involved in sensory discrimination of nutritious foods or in motor control of proboscis extension. Rather, they seem to be the site of a motivational state of low energy, and drive homeostatic feeding. This is supported by the fact that a subpopulation of these neurons responds to starvation with an increase in calcium levels, and that these neurons seem to regulate Dilp2 secretion from the insulin-producing cells. If true and novel these findings provide an important advance in our understanding of the neuronal control of motivational states, particularly given the current interest in starvation responses in the *Drosophila* brain.

The experiments performed clearly and thoroughly validate the authors' conclusions that a small set of 'Taotie' neurons represents a master regulator of feeding behaviour in response to starvation in *Drosophila*. The behavioural phenotypes are strong and clear, and the data are presented in a clear and coherent manner. The paper is organised in a logical manner, starting with behavioural characterisation and then proceeding to anatomical specificity and finally concluding with interactions with other known appetite-controlling neurons. Furthermore, the supplementary materials provide data that is highly relevant for the conclusions of the paper, and supports the data presented in the main figures, without providing unnecessary detail or extra interpretations. Unfortunately the paper could be better written. Avoiding overstatements, and exaggerated claims in relation to the impact on vertebrate biology and human health. Putting better into the context of what is known in invertebrates, adding the dimension of what is known about the regulation of other nutrients (like proteins) and also comparing it to the circuit details in vertebrates (AgRP and POMC neurons) would in my opinion provide a better introduction and discussion. Also a thorough improvement of the English would help.

My main concern with this manuscript is whether the neurons identified in this study are distinct from those identified by Dus et al (Neuron, 2015), which express the neuropeptide Dh44 and control the selection of nutritive over non-nutritive sugars in starved flies. Both Taotie neurons and Dh44 neurons have their cell bodies in the pars intercerebralis, and have very similar projection patterns in the prow. The behavioural phenotypes of the two lines are somewhat distinct: while silencing Taotie neurons abolishes the increase in food intake normally seen upon starvation, silencing of Dh44 neurons abolishes the preference of starved flies for nutritive sugars, without affecting overall food intake. Furthermore, activation of Dh44 neurons (using NaChBac) does not cause sated flies to behave as if they were starved. However, these differences could be due to differences in the experimental protocols. Therefore, since both neuron types control some aspects of feeding behaviour, it seems essential to provide conclusive evidence of whether these are indeed distinct populations of neurons, both for the novelty of this study and for the field as a whole. I would suggest one or more of the following experiments: immunostaining using an antibody against Dh44 together with staining of the Taotie neurons; colabelling of Taotie-Gal4 and Dh44-Gal4; Dh44 knockdown using RNAi in Taotie neurons. In any case it is essential that the authors address this question. If the Taotie neurons and Dh44 neurons the authors need to put their findings in the context of the phenotypes observed when manipulating Dh44 neurons as well as the imaging experiments conducted in that study and explain how their study advances the field beyond what was described by Dus and colleagues. If the neurons are not the same the authors need to look at the relation of the Taotie neurons with the Dh44 neurons and discuss how the results from both studies could relate to each other.

I applaud the authors' thoroughness in their characterisation of the behavioural phenotypes evoked by manipulations of Taotie neurons. They set an example for the field with the thoroughness and quality of this work. However, I am concerned that perhaps the manuscript may

be somewhat over-long for a general reader. In particular, some of the figures could be condensed, since some of the panels present information that is somewhat redundant with other panels. I would suggest that figure 5, while it does add to the content of the study, is not a primary finding and is not discussed extensively in the text, and so may be moved to the supplementary figures. The authors may consider this also for some specific panels, for example 1e, 4e, 6b, 6e and 8c.

As said the authors' language in the introduction and discussion does at times tend towards overstating. In particular, I am not convinced about terming the pars intercerebralis generally, or the Taotie neurons specifically, as a 'master regulator' of feeding behaviour. It is known that there are neurons elsewhere in the brain that influence food intake, and besides it seems that these neurons represent the internal state of hunger. In fact, these neurons may influence distinct aspects of behaviour in addition to feeding, such as memory, which the authors did not address in this study. In any case, I would suggest using less decisive language in describing these neurons, essentially avoiding the word 'master'.

Given these reservations I am afraid I cannot recommend this study for publication in this form. However, if the authors are able to address all the above stated major concerns and the appended minor concerns, this study could be suitable for publication in Nature Communications.

Minor comments:

The title contains a typo: 'mater' should presumably be 'master'

In the abstract, the authors mention that the activity of these neurons 'induces changes in feeding motivation, regardless of physiological status'. This phrasing is unclear given that there is no activation phenotype in starved animals. I would therefore suggest to rephrase this, for example as 'induces bidirectional changes in feeding motivation'.

In the abstract, the authors claim that 'these "gluttony" neurons coordinate feeding behavior and physiological homeostasis by controlling insulin secretion'. This claim is not shown by the data presented here; while the activity of these neurons does control insulin secretion, and insulin secretion does control feeding behavior, there is no direct evidence that the effect of these neurons on feeding behavior goes through insulin. To show this would require some genetic or neuronal epistasis. I would suggest rephrasing this as 'neurons control the secretion of insulin, a known regulator of feeding behavior and physiological homeostasis', or something similar. I would also avoid the use of the word "gluttony". Further, the phrase 'in both satiated and hungry flies' is not clear; I would suggest modifying this to 'according to hunger state' or something similar.

Page 2: 'Drosophila melanogaster represents to be' is not correct; the 'to be' should be removed'

Page 2: it is not clear what the authors mean by 'highly conserved metabolic homeostasis'. Could they be more specific, for example conserved molecular pathways, or conserved principles controlling feeding behavior?

Page 2: 'For example, studies of the dopamine and its receptor neurons have revealed that an extremely complicated circuitry exists'. To me, this seems rather hyperbolic; I would rather simply state that distinct dopaminergic neurons in different brain areas act at multiple control points.

Page 2: There have recently been some nice studies using automated methods to study feeding behaviour at high resolution (e.g. FLIC and flyPAD). The authors should use these descriptions of feeding when describing the process of feeding (motivational, sensory, motor).

Page 3: 'presenting the internal energy status' should be 'representing'

Page 3: I don't think this study says anything about evolutionary aspects of circuit function. I would strongly urge the authors to remove the last two hyperbolic sentences at the end of the page.

I would mention or show the elav-Gal80 data very early in the manuscript as they make the important point that the labelled cells driving the behavioural effect are neurons.

The data in figure 1C is described but no statistics are presented. I would suggest that the authors should use a Fisher's exact or Chi square test to check whether the feeding phenotype by this measure is significantly different from the control lines.

The data presented in figure 1f and g clearly shows that activation of this line evokes a preference for the food-containing area of the dish. However, I am not convinced that this represents 'active food-seeking' behavior, as the authors claim. Rather, it seems to be a place preference for food-containing areas.

It is interesting to see that the phenotypes evoked by long-term activation of Taotie neurons revert back to normal following cessation of the activation. It would be informative to see whether this reversion is due to a compensatory decrease in food intake below baseline levels; to test this, the authors could look at food intake, for example, 1 day following the cessation of neuronal activation.

Figure 3d - the authors should perform a statistical comparison of the groups shown in this panel.

Page 8 - Previous studies have shown that flies use distinct behavioral and neuronal mechanisms to control feeding behavior in response to different periods of starvation. I would suggest that the authors could mention these previous studies, particularly Itskov et al, 2014 and Inagaki et al, 2012.

Page 8 - the authors write 'the difference between the activated group and the starved group became progressively smaller, suggesting that in the activated flies, the impact induced by Taotie-Gal4 activation was absorbed by the assumed hunger mechanism'. This description is unclear and not well explained. I would suggest modifying this sentence to read 'the difference between the activated group and the starved group became progressively smaller with increased starvation time, suggesting that Taotie-Gal4 activation mimicked the effect of 12h starvation'.

Page 9 - 'After activation for 30 minutes or more in the absence of food, Taotie>CsChrimson flies exhibited strong food intake in the subsequent feeding assay, without further activation required (Fig. 6a,b).' It is not completely clear from this explanation that the activation is stopped before the assay begins; I suggest rewriting the sentence slightly to make this point clearer.

The only phenotype which does not fit the fact that the neuronal manipulation mimics starvation is the fact that the flies do not ingest bitter food. It has been clearly described that starvation makes flies not avoid bitter food anymore. The authors should put their findings in the context of these findings. Maybe the induced starvation state is not strong enough to show that effect or the bitter content of food is too high?

Page 9 - 'strongly argues against that these neurons locate in the input..' is poor grammar. I would suggest 'strongly suggests that these neurons are not located in the input or output pathways of feeding control, but have a central position...'

The experiment in which the authors briefly activate the Taotie neurons followed by a delay period is very informative as to their temporal role. In this experiment, there is no food present during the delay period, which is an important factor. I would suggest that we could learn more about the sustained phenotype by also repeating the experiment with food present in the delay period. In

this way, 1) the experiment could be extended to longer time points, which is currently limited by when the control group attains the same feeding level as the activated flies due to starvation; and b) it is possible that a transient activation activates a long-lasting motivational state that is only reversed when food is presented. If this is the case, then the presence of food during the delay period would abolish the persistent effect following activation. Either way, this experiment would give an interesting result.

Page 9 - the subesophageal ganglion has been renamed as the subesophageal zone, according to Ito et al, 2014.

Page 10 - the imaging result showing that these neurons are activated following starvation is very nice. However, in the description of the result, the authors should be more specific about what the data is. Rather than 'the PI region of hungry flies exhibited elevated reporter fluorescence, indicating that these neurons are highly activated when flies are hungry', they should specify that they are using a reporter of intracellular calcium levels, for example 'the neurons in the PI region exhibited increased GCaMP fluorescence following starvation, indicating that these neurons are activated, with high intracellular calcium levels, when flies are hungry'. Likewise, the authors should indicate that 'feeding hungry flies for 1 hour reduced the calcium level of these neurons to the level of satiated flies', rather than the 'activities of these neurons'.

The authors indicate that other neurotransmitters and neuropeptide systems are unlikely to interface with Taotie neurons based on their previous behavioural screen. To make this claim, the authors need to show the results of this behavioural screen. Also I am not sure that you can take these negative results as strong evidence. Especially not that this shows that the Taotie neurons are part of a high-level neuronal circuit. Absence of evidence is not evidence of absence.

Page 14 - the authors note that 'In the periphery, both olfactory and gustatory inputs related to feeding are affected by hunger'. The authors should cite references 17 and 18 in addition to 3 and 45 to support this point.

Page 16/17 - the authors suggest that 'as the serotonin neurons project broadly in the brain, they are presumably more suitable for modulating arousal globally'. I do not agree with this statement, and there is no evidence to suggest that the serotonergic neurons that evoke hunger are generally modulating arousal. Further, I do not quite understand the point the authors are trying to make here. In fact, it seems to me that the behavioural phenotypes of Taotie neurons are very similar to those demonstrated by Albin et al, 2015 for a subset of serotonergic neurons. Thus, these Taotie neurons are not the first demonstration of neurons in *Drosophila* that represent a motivational state of hunger and modulate different behaviours accordingly. However, this does not detract from the importance of this study but the authors should still discuss this work at the same level as theirs. Further, the authors mention that 'activating all serotonergic neurons with Tph-GAL4 in satiated flies did not elevate their feeding motivation'. This is true and was in fact shown by Albin et al, 2015. This indicates that global manipulations obscure the phenotypes observed upon manipulations of small subsets of neurons, and does not in any way invalidate the findings of Albin et al.

Page 18 - 'To disseminate hunger-driven behavior...' - I suppose this should be 'To dissect hunger-driven behavior...'.
Figure 8b,d,e - I presume that the scale is normalised fluorescence intensity; this should be defined in the figure or the figure legend.

The authors should avoid the term insulin and use the correct nomenclature for the dIlp peptides.

Again I would be very careful in making any conclusions on obesity in humans as well as interventions for therapies using these studies. It is unnecessary and too speculative.

Reviewer #3 (Remarks to the Author)

This manuscript addresses the fundamental question of energy balance and the control of feeding behavior in flies. A group of neurons is identified using the Taotie-gal4 driver line that promotes feeding behavior upon neural activation in satiated flies. Stimulation of these neurons using the Taotie driver induces an obese phenotype due to overfeeding, while their inhibition reduces feeding even after starvation. Taotie-activated flies present similar appetite as 12hr-starved animals, with identical food preference. Activation of Taotie neurons induces Dilp retention in insulin-producing cells, suggesting that the Taotie neurons could modulate feeding behavior through a general control of insulin signaling.

Experiments are well crafted and in many instances support the conclusions of the paper. The message is significant and should contribute to a better knowledge of the links between homeostatic/physiological hunger and neurological activation of feeding. However, the manuscript does not address the crucial question of homeostatic signals upstream of the taotie neurons, despite the possibility of an obvious molecular link. In conclusion several major and minor remarks need to be addressed before publication.

Major comments:

1- The Taotie-Gal4 line contains the promoter region of the gustatory receptor Gr28b.b gene. This suggests that this gustatory receptor could participate in central nutrient sensing by the Taotie neurons (See Miyamoto et al. 2012 for the case of Gr43a). Why is this not experimentally tested, nor mentioned in the paper? This is a crucial point that needs to be addressed, since it could provide a key missing link between the nutritional status of the animal and the activity of the Taotie neurons.

2- It is not clear why taotie neuron activation takes 30 min. to induce feeding. This seems desperately long for a circuitry that would transduce physiological hunger into neurological hunger in physiological conditions.

3- There is a hole in the experimental logic describing the link between the Taotie neurons and the IPCs. In fed animals, Taotie neuron activation blocks insulin release. However, this cannot be causal to the observed feeding increase, since blocking IPC activity using Kir2 or Shits is not sufficient to induce feeding in these conditions (see Suppl. Fig. b,c).

Minor comments:

1- In many instances, redundant data and/or control experiments are presented in independent panels, which should be moved to suppl. Material. For instance: Fig. 1a,b,e,f,j; Fig. 4b,d,h; Fig. 6b,e.

2- Why using colorimetric quantifications of a food dye on whole flies where a cafe assay would be more quantitative?

3- Interpretation of results in Fig. 1d seems misleading: starvation enhances the effect of taotie>trpA1 by three fold, meaning that taotie neuron activation is only part of the nutritional response, but not sufficient for full response.

4- A convincing test that activation of taotie neurons increase food search behavior would be to look for general increase of motility associated to food search. Are animals more motile upon taotie neuron activation in the absence of food?

5- In the absence of more data concerning the effect of Taotie neuron activation on obesity, it seems useless to compare this with the obese syndrome in human and its reversibility. Are Taotie-activated obese flies presenting some characteristics of the obesity syndrome (insulin resistance, general inflammatory response etc..)?

6- In fig. 6d, the long-lasting motivation for feeding induced by taotie neuron activation could simply be due to the non-physiological activation procedure. Therefore, the proposed conclusion seems inappropriate.

7- The authors should avoid emphatic terms like "master feeding center".

8- Fig 7 should present a magnification of the PI cluster showing individual neurons for clarity.

9- The authors should comply to the nomenclature, i.e. insulin-producing cells are called IPCs, not DPMNs.

Reviewer #1 (Remarks to the Author):

This manuscript describes the identification of a subset of neurons within the adult brain, called Taotie neurons, which control feeding behavior in *Drosophila*. Specifically, the authors demonstrate that these neurons regulate the hunger/satiety signal such that activation of these neurons causes flies to continuously feed despite being fully fed leading to obese phenotypes. The authors further demonstrate that these neurons are not directly controlled by insulin signaling but do coordinate feeding and physiological homeostasis by regulating insulin secretion.

Overall, this is an extremely interesting area of research aimed at understanding the neural networks that regulate appetite and feeding behavior. It has important implications in our understanding and potential treatment of obesity and obesity associated disorders. While several other studies have previously demonstrated a role for the pars intercerebralis in feeding behavior, Taotie neurons appear to overlap with PI but have distinct functions. Similarly, other studies have shown roles for various neurotransmitters and neuropeptides regulating hunger induced behavior after starvation, but this also appears distinct from the ravenous feeding behavior induced by Taotie neuron activation. Thus, there are several novel aspects to this manuscript. However, in some ways the manuscript is a careful analysis of a group of neurons identified in a GAL4 screen. The authors provide very little data regarding the expression pattern of Taotie-GAL4 and no information regarding what signaling pathways are active/required in these neurons, which is the key question.

Appropriate use of statistics and treatment of uncertainties: Yes

We thank the reviewer for the encouraging comments. In the revised manuscript, we have included a full characterization of the *Taotie-GAL4* expression, including their projection patterns, the pre- and post-synaptic sites, and co-localization analysis with insulin producing cells and Dh44 neurons. We also added new results from investigating physiological factors that lead to activation of Taotie neurons and intracellular signaling pathways that are required for the feeding behavior evoked by Taotie neurons.

Suggested improvements:

The authors examine the effect of activating Taotie neurons on food preferences. Given the links to obesity and diabetes, it would have been useful to determine how these flies respond to high sugar and high fat diets. Are these flies more sensitive to these manipulations?

We are grateful for the reviewer's suggestion. To address this question, we conducted additional experiments. Our data clearly show that chronic activation of Taotie neurons elevates triglyceride and glucose level compared with a genetic control on high sugar diet, suggesting that energy

homeostasis is further skewed under high energy diet treatment (Supplementary Fig. 2c,d). Next, we tested whether blocking the Taotie neurons would attenuate obese phenotypes on high sugar diet. We found that inactivation of Taotie neurons effectively reduces levels of glucose and triglyceride induced by high sugar diet, but not those by regular food (Fig. 3e,f), as had been shown in human previously.

We did not test flies with high fat diet because flies were prone to stick to the surface of food with high fat (regular food +30% coconut oil) resulting in flies unhealthy for behavior analysis.

The authors demonstrate that the obese phenotype is reversible but only examine one timepoint. This analysis should be expanded.

We agree with the reviewer, and now provide additional data with more time points in the reversal period. All the obese phenotypes are gradually reverted back to normal states within 4 days after withdrawing light stimulation (Fig. 2 and Supplementary Fig. 2a). Interestingly, we found lower food content in these obese flies compared to genetic control flies in the reversal period, suggesting that reversed obese phenotypes are due to a compensatory reduction of food intake (Supplementary Fig. 2b).

The authors examine the relationship of Taotie activation with insulin signaling. Is there any link to the fly equivalent of leptin/leptin R (e.g. *Upd1/Dome*)?

We thank the reviewer for raising this interesting point. In *Drosophila*, cytokines *Upd1*, *Upd2* and *Upd3* bind the same receptor, *Domeless* (*Dome*) which is the homolog of GP130 in vertebrates. Previous studies indicated that *Upd1* plays a crucial role in stem cell self-renewal and differentiation, while *Upd2*, the fat-body-expressed cytokine, acts as the functional homolog of mammalian leptin, which transfers satiated signals to the central neural system. Cytokine *Upd3* is involved in gut infection and homeostasis.

Although *Upd2* is the leptin homolog in *Drosophila*, deletion of *Upd2* in flies showed little effect on their feeding behavior. As anti-leptin antibodies are not available, we performed RT-PCR to examine the mRNA level of *Upd2* in flies with chronic activation of Taotie neurons, in order to test the relationship of Taotie activation with leptin system. We found that *Upd2* mRNA was downregulated dramatically compared with the controls (Supplementary Fig. 3d). This is in agreement with our hypothesis that activation of Taotie neurons represents endogenous neurological hunger states. The hunger signals induced by Taotie neuron activation suppress multiple satiation

signals, including *Upd2*, to mount an effective feeding actions in order to coordinate feeding with physiological homeostasis. It would be interesting to identify potential cross-talks between Taotie neurons and metabolic tissues, however mechanisms underlying this process are currently unknown. It is reported that human leptin rescues the phenotypes of *Upd2* mutant flies. Additionally, both *Upd2* and human leptin engage the receptor *Dome* to activate JAK/STAT pathway. In our experiments with flies of induced obesity, the mRNA levels of *Domeless* displayed little changes, although the *Upd2* signal was downregulated (Supplementary Fig. 3d). A simple explanation for this observation might be that the expression levels of *Domeless* are not under direct regulation of the *Upd2* signal. Alternatively, it is also possible that *Domeless* expression levels are regulated by the signal strength of the corresponding pathways of the three ligands (*Upd1*, *Upd2* and *Upd3*), thus its expression is the net result of the activities of all three pathways. The combined effects of downregulation of *Upd2* (Supplementary Fig. 3d) and upregulation of *Upd3* (due to inflammatory response, Supplementary Fig. 3c) in Taotie activated obese flies would thereby result in an unchanged overall expression level of *Domeless*. Further experiments would be required to distinguish between the two possibilities. For now, we could not draw a direct conclusion in terms of whether the induced obesity by Taotie neuron activation influences the leptin receptor under this circumstance.

The authors should consider using RNAi approaches to examine the role of known signaling pathways involved in feeding/appetite regulation using their Taotie-GAL4 driver.

We thank the reviewer#2 for this excellent suggestion. We have now incorporated the results of an RNAi screen using the *Taotie-GAL4* driver. In addition to the RNAi approach, we utilized other strategies to explore potential signaling pathways involved in overfeeding behavior mediated by Taotie neurons at both molecular and cellular levels.

First, our results suggest that Taotie neurons evoke feeding behavior through a neuropeptide dependent pathway. We conducted a behavioral RNAi screen covering six neurotransmitters, forty neuropeptides and corresponding synthesis enzymes and transporters to knockdown their expression levels in Taotie neurons. The RNAi stocks were collected from the Bloomington Stock Center, VDRC, and THU stock center. However, we did not observe significant effects of knocking-down single neuromodulator on the feeding behavior evoked by Taotie neurons. There are at least have two possibilities that account for this result. One is due to relatively low efficiency of the RNAi

method, which has been observed in multiple cases in other behavioral paradigms in our lab. Another possibility is that the process inside of Taotie neurons leading to evoking feeding behavior employs multiple neuromodulators to work redundantly.

Interestingly, from this RNAi screen, we found that disrupting the processing of neuropeptides via knocking down proprotein convertase, *Amon*, in Taotie neurons decreased feeding behavior evoked by Taotie neuron activation under satiated conditions (Supplementary Fig. 13a). Furthermore, over-expression of full-length *Amon* in Taotie neurons increased food consumption (when fed *ad libitum*), while knocking down *Amon* in Taotie neurons inhibited food intake in starved flies (Supplementary Fig. 13b,c). Together, these data demonstrate that Taotie neurons evoke feeding behavior at least partially through a neuropeptide dependent pathway.

Second, our new data showed that hemolymph glycemia regulates the activity of Taotie neurons. In order to establish causal link between internal nutritional status of the animal and the activation of Taotie neurons during hunger response, we manipulated hemolymph glycemia via diet or drugs and measured the calcium influx in Taotie neurons in responding to different treatments (Fig. 6g,h). Flies fed with non-nutritious sugar, protein-rich food, or hypoglycemic drugs led to hypoglycemia, also caused high activities in Taotie neurons, while all treatments of nutritional sugars that lead to elevated glycemia, maintained with basal level activities in Taotie neurons (Fig. 6g,h), demonstrated that neural activity of Taotie neurons were regulated by hemolymph glycemia levels. This conclusion is further confirmed by our re-feeding experiment, as caloric sugars effectively suppressed neural activity of Taotie neurons, while non-caloric sugars and proteins-rich food did not have suppressive effects (Supplementary Fig. 14). Thus, we propose that Taotie neurons serve as a mediator connecting physiological status with neurological processes by receiving signals of hemolymph glycemia and commanding feeding behavior coordinately.

Third, to gain further insights into the epistatic relationship of Taotie neurons with other feeding control neurons, we sought to investigate the interactions between positive feeding signals presented by Taotie neurons and the negative signals by known satiated signals (*AstA*, *Dilp2* and *Gr43a*) in fruit flies. When co-activating Taotie neurons with these satiation promoting neurons, the feeding behavior evoked by activating Taotie neurons were suppressed, but not diminished completely (Supplementary Fig. 19), suggesting that activation of Taotie neurons promotes feeding behavior through a circuit mechanism that counteracts with satiated pathways in flies.

Fourth, we used pre-synaptic marker *UAS-Syt-GFP* and post-synaptic marker *UAS-DenMark* to visualize the projection patterns and synaptic distributions of Taotie neurons in the brain (Supplementary Fig. 11). Confocal imaging data showed that the post-synaptic sites locate abundantly throughout the Taotie neurons, suggesting that Taotie neurons receive multiple inputs. Results with the pre-synaptic marker showed that Taotie neurons project into the subesophageal zone, in agreement with the supposed role of Taotie neurons to control feeding behavior bi-directionally. Together, our imaging data provide important structural information for a better understanding of the functions of Taotie neurons.

Reviewer #2 (Remarks to the Author):

To optimise survival animals must match their energy intake to their current needs at any point in time. To do this, the brain must regulate feeding behaviour according to the animal's metabolic state. We have some insight into neurons performing some components of this function in mammals and in other organisms, but a circuit level understanding is missing. This study provides an important addition to our understanding of the neuronal control of energy homeostasis and feeding control in *Drosophila*, one of the organisms in which such an understanding could be achieved within a reasonable timeframe.

The authors set out to identify novel neuronal components controlling food intake in *Drosophila*. To do this, they conducted an activation screen to find neurons that increase food intake when activated, and identify one line, Taotie-Gal4, that drives food intake. The authors use bidirectional manipulations of neuronal activity to show that these neurons are both necessary and sufficient for the feeding response to starvation of up to 12 hours. They perform a thorough characterisation of the feeding phenotypes evoked by this line, showing that these neurons control various feeding-related behaviours in a similar way to starvation. Importantly, the authors show that these neurons do not seem to be involved in sensory discrimination of nutritious foods or in motor control of proboscis extension. Rather, they seem to be the site of a motivational state of low energy, and drive homeostatic feeding. This is supported by the fact that a subpopulation of these neurons responds to starvation with an increase in calcium levels, and that these neurons seem to regulate Dilp2 secretion from the insulin-producing cells. If true and novel these findings provide an important advance in our understanding of the neuronal control of motivational states, particularly given the current interest in starvation responses in the *Drosophila* brain.

The experiments performed clearly and thoroughly validate the authors' conclusions that a small set of 'Taotie' neurons represents a master regulator of feeding behaviour in response to starvation in *Drosophila*. The behavioural phenotypes are strong and clear, and the data are presented in a clear and coherent manner. The paper is organised in a logical manner, starting with behavioural characterisation and then proceeding to anatomical specificity and finally concluding with interactions with other known appetite-controlling neurons. Furthermore, the supplementary materials provide data that is highly relevant for the conclusions of the paper, and supports the data presented in the main figures, without providing unnecessary detail or extra interpretations. Unfortunately the paper could be better written. Avoiding overstatements, and exaggerated claims in relation to the impact on vertebrate biology and human health. Putting better into the context of what is known in invertebrates, adding the dimension of what is known about the regulation of other nutrients (like proteins) and also comparing it to the circuit details in vertebrates (AgRP and POMC neurons) would in my opinion provide a better introduction and discussion. Also a thorough improvement of the English would help.

We thank the reviewer#2 for all constructive suggestions. In our revised version, we have changed all statements regarding the impact of our findings on our understanding of vertebrate biology and human health. We also paid more attention to discussing our findings in relation to the neurological and physiological studies in invertebrates. We have now added more data for a comparison of the effects of sugar diet and protein diet on activation of Taotie neurons. We also compared the Taotie neurons with the AgRP neurons, which play important roles in control feeding and energy homeostasis in vertebrates.

My main concern with this manuscript is whether the neurons identified in this study are distinct from those identified by Dus et al (Neuron, 2015), which express the neuropeptide Dh44 and control the selection of nutritive over non-nutritive sugars in starved flies. Both Taotie neurons and Dh44 neurons have their cell bodies in the pars intercerebralis, and have very similar projection patterns in the prow. The behavioural phenotypes of the two lines are somewhat distinct: while silencing Taotie neurons abolishes the increase in food intake normally seen upon starvation, silencing of Dh44 neurons abolishes the preference of starved flies for nutritive sugars, without affecting overall food intake. Furthermore, activation of Dh44 neurons (using NaChBac) does not cause sated flies to behave as if they were starved. However, these differences could be due to differences in the experimental protocols. Therefore, since both neuron types control some aspects of feeding behaviour, it seems essential to provide conclusive evidence of whether these are indeed distinct populations of neurons, both for the novelty of this study and for the field as a whole. I would suggest one or more of the following experiments: immunostaining using an antibody against Dh44 together with staining of the Taotie neurons; colabelling of Taotie-Gal4 and Dh44-Gal4; Dh44 knockdown using RNAi in Taotie neurons. In any case it is essential that the authors address this question. If the Taotie neurons and Dh44 neurons the authors need to put their findings in the context of the phenotypes observed when manipulating Dh44 neurons as well as the imaging experiments conducted in that study and explain how their study advances the field beyond what was described by Dus and colleagues. If the neurons are not the same the authors need to look at the relation of the Taotie neurons with the Dh44 neurons and discuss how the results from both studies could relate to each other.

We thank the reviewer for raising these important points. We agree that distinguishing the neurons labeled by *Taotie-GAL4* from those labeled by Dh44 is important for future studies on both feeding motivation and satiation mechanisms in *Drosophila*. In this revision, we provided additional evidences to indicate that Taotie neurons and Dh44 neurons are indeed two distinct populations of neurons, which control different aspects of feeding in *Drosophila*.

First, for a direct comparison with Taotie neurons, we conducted a feeding assay in flies with

Dh44 neurons activated by optogenetics. Unlike activation of Taotie neuron, activation of Dh44 neurons did not affect food intake in satiated flies in our paradigm (Supplementary Fig. 18g). This result is consistent with the previous study by *Dus* and colleagues, strongly suggesting that manipulating the activity of Dh44 neurons do not generate a similar effect as manipulating Taotie neurons.

Second, as both of Dh44 and Taotie neurons are located in the PI region, we performed two color co-labeling experiment with Taotie>GFP and Dh44>RFP to examine the spatial relationships of the two populations of neurons (Supplementary Fig. 18a,b and Supplementary Video 2). Through analyzing every single sections of multiple imaging stacks, we concluded that Taotie neurons are rarely co-labeled with Dh44 neurons (about 1-2 overlapped neurons per brain), suggesting that Taotie neurons and Dh44 neurons are two distinct populations of neurons.

Third, data from our genetic manipulation also support the conclusion derived from anatomical studies. We combined *LexAop-GAL80* with *Taotie-GAL4*, *Dh44-LexA*, and *UAS-effectors* (*UAS-GFP*, *UAS-ReaChR*, or *UAS-Kir2.1*) to visualize or manipulate only those Taotie neurons that are not Dh44 positive (Supplementary Fig. 18c,d). From behavioral tests, we found that excluding Dh44 positive neurons from the Taotie population did not change the feeding phenotypes of both activation of Taotie neurons in satiated flies and inactivation of Taotie neurons in starved flies (Supplementary Fig. 18e-f). Together, our results demonstrate that the Taotie positive and Dh44 negative neurons are both necessary and sufficient to evoke feeding in satiated flies and to suppress feeding in hungry flies. Therefore, Taotie neurons, but not the Dh44 neurons, are responsible to the phenotypes described in our experiments.

Fourth, as the glucose-sensing Dh44 neurons function as a post-ingestive nutrient sensor through detecting the increased hemolymph glycemia after feeding, we wondered the behavioral phenotype of co-activation of both Taotie neuron and Dh44 neurons. To our surprise, co-activation of both neurons did not alter the amount of food intake evoked by activation of Taotie neurons alone (Supplementary Fig. 18g). This result is different from other epistatic results where co-activation Taotie neurons and other neurons related to satiety signals decrease the overfeeding phenotype of Taotie neuron activation (Supplementary Fig. 19). Therefore, Taotie neurons and Dh44 neurons regulate different aspects of feeding process.

Together, with additional evidences, we showed that Taotie neurons and Dh44 neurons are both

anatomically and functionally distinct. Because both neurons control different aspect of feeding and both are located in the PI region, this nevertheless highlights the importance of the PI region as a common area for both neuroendocrine functions and feeding control.

I applaud the authors' thoroughness in their characterisation of the behavioural phenotypes evoked by manipulations of Taotie neurons. They set an example for the field with the thoroughness and quality of this work. However, I am concerned that perhaps the manuscript may be somewhat over-long for a general reader. In particular, some of the figures could be condensed, since some of the panels present information that is somewhat redundant with other panels. I would suggest that figure 5, while it does add to the content of the study, is not a primary finding and is not discussed extensively in the text, and so may be moved to the supplementary figures. The authors may consider this also for some specific panels, for example 1e, 4e, 6b, 6e and 8c.

We thank the reviewer#2 for appreciating our efforts related to the behavioral characterization of the overfeeding phenotypes evoked by activation of Taotie neurons. We took the reviewer's suggestion to make the manuscript concise. In the new version, we removed some redundant panels (Fig. 4e, 6b, 6e, and 8c) into the supplementary figures. We also moved figure 5 into supplementary figures (now Supplementary Fig. 4) as suggested.

As said the authors' language in the introduction and discussion does at times tend towards overstating. In particular, I am not convinced about terming the pars intercerebralis generally, or the Taotie neurons specifically, as a 'master regulator' of feeding behaviour. It is known that there are neurons elsewhere in the brain that influence food intake, and besides it seems that these neurons represent the internal state of hunger. In fact, these neurons may influence distinct aspects of behaviour in addition to feeding, such as memory, which the authors did not address in this study. In any case, I would suggest using less decisive language in describing these neurons, essentially avoiding the word 'master'.

In this new version, we removed decisive sentences and overstating phrases, including 'master regulator', from the title and multiple places in the text. However, we strived to find proper words to describe the importance and uniqueness of the Taotie neurons. From a behavioral point of view, this is a new set of neurons promoting overfeeding in satiated flies (Supplementary Fig. 20). Interestingly, a subset of Taotie neurons form a cluster in a defined brain region. Also, Taotie activation is part of the hunger process, during which Taotie activation represents the hunger state and feeding motivation. We suggest that the hunger-derived motivation for feeding is distinct from food preference of starved flies, as these are two processes occurring at different levels of feeding controls.

Given these reservations I am afraid I cannot recommend this study for publication in this form. However, if the authors are able to address all the above stated major concerns and the appended minor concerns, this study could be suitable for publication in Nature Communications.

Minor comments: The title contains a typo: 'mater' should presumably be 'master'

We now change our title into 'Identification of a feeding circuit for physiological homeostasis in *Drosophila*', hopefully to good effect.

In the abstract, the authors mention that the activity of these neurons 'induces changes in feeding motivation, regardless of physiological status'. This phrasing is unclear given that there is no activation phenotype in starved animals. I would therefore suggest to rephrase this, for example as 'induces bidirectional changes in feeding motivation'. We thank the reviewer for this excellent suggestion, and have changed the passage to 'induces bidirectional changes in feeding motivation'.

In the abstract, the authors claim that 'these "gluttony" neurons coordinate feeding behavior and physiological homeostasis by controlling insulin secretion'. This claim is not shown by the data presented here; while the activity of these neurons does control insulin secretion, and insulin secretion does control feeding behavior, there is no direct evidence that the effect of these neurons on feeding behavior goes through insulin. To show this would require some genetic or neuronal epistasis.

We agree with the reviewer that the effect evoked by activation of Taotie neurons does not go through insulin exclusively. The activation of Taotie neurons likely results in two relevant events of feeding control, one occurs in activating the feeding circuit, the other is to suppress the inhibitory effects of satiation.

We conducted neural epistatic experiment to co-activate Taotie neurons and Dilp2 cells in satiated flies, and found that the overfeeding phenotype evoked by activation of Taotie neurons was greatly decreased under this circumstance (Supplementary Fig. 19b). Thus, these results suggested that the normal activity of Taotie neurons may serve to keep the activities of IPC cells at a low level, since an otherwise high level of activity in IPC induces suppression of feeding (Supplementary Fig. 16a). We have changed the wording in the revised version of our manuscript accordingly.

I would suggest rephrasing this as 'neurons control the secretion of insulin, a known regulator of feeding behavior and physiological homeostasis', or something similar.

We thank the reviewer and took the sentence into the abstract, where it now states 'Importantly, these appetitive-enhancing neurons control the secretion of insulin, a known regulator of feeding

behavior and physiological homeostasis according to hunger state'.

I would also avoid the use of the word "gluttony". Further, the phrase 'in both satiated and hungry flies' is not clear; I would suggest modifying this to 'according to hunger state' or something similar.

We eliminated 'gluttony' from the text, and also re-wrote the sentence according to the reviewer's suggestion.

Page 2: '*Drosophila melanogaster* represents to be' is not correct; the 'to be' should be removed'

Thanks for reviewer's help to fix our grammar mistakes. We have amended the sentence accordingly.

Page 2: it is not clear what the authors mean by 'highly conserved metabolic homeostasis'. Could they be more specific, for example conserved molecular pathways, or conserved principles controlling feeding behavior?

We re-wrote this sentence, now it reads as: '*Drosophila melanogaster* represents an excellent model for investigating the essential molecular and neuronal mechanisms underlying feeding behavior, for both wide availability of powerful genetic tools and highly conserved molecular pathways from flies to mammals in metabolic homeostasis'.

Page 2: 'For example, studies of the dopamine and its receptor neurons have revealed that an extremely complicated circuitry exists'. To me, this seems rather hyperbolic; I would rather simply state that distinct dopaminergic neurons in different brain areas act at multiple control points.

We agree with the reviewer's suggestion, and re-wrote the sentence.

Page 2: There have recently been some nice studies using automated methods to study feeding behaviour at high resolution (e.g. FLIC and flyPAD). The authors should use these descriptions of feeding when describing the process of feeding (motivational, sensory, motor).

We cited these works in the introduction, and described feeding process in aspects of motivation, sensory input, and motor in the appropriate passages in this version of manuscript.

Page 3: 'presenting the internal energy status' should be 'representing'.

We modified all affected sentences as suggested.

Page 3: I don't think this study says anything about evolutionary aspects of circuit function. I would strongly urge the authors to remove the last two hyperbolic sentences at the end of the page.

We removed these two sentences in the new version as suggested.

I would mention or show the elav-Gal80 data very early in the manuscript as they make the important point that the labelled cells driving the behavioural effect are neurons.

It would be nice to show the *elav-GAL80* results very early in the manuscript. However, this would generate a new supplementary figure, as this anatomical result does not combine well with the data of behavioral characterization. We hope that our strategy of screening GAL4 lines labeled different population of neurons in flies' brain via activating neurons by thermogenetics as well as behavioral characterization by optogenetics would help to convince the reader that these neurons are indeed the targets of our study.

The data in figure 1C is described but no statistics are presented. I would suggest that the authors should use a Fisher's exact or Chi square test to check whether the feeding phenotype by this measure is significantly different from the control lines.

We have now conducted statistical tests for Figure 1C, and added the result in the figure legend of Figure 1C.

The data presented in figure 1f and g clearly shows that activation of this line evokes a preference for the food-containing area of the dish. However, I am not convinced that this represents 'active food-seeking' behavior, as the authors claim. Rather, it seems to be a place preference for food-containing areas.

We agree with the reviewer that the behavioral consequence depicted in previous Figure 1f and g is also used for quantifying food/place preference. However, as shown in Figure 1f, the hungry flies gather at food-containing area, no matter of their original positions. Before they stop at the food patch, the dynamic process of searching and walking is an active food-seeking behavior. We respectfully suggest that the place preference for food-containing areas does not conflict with or exclude active food-seeking behavior.

In this revision, to further demonstrate that the flies with activated Taotie neurons actively seek food without the reward of reaching food, we added a new set of experiments to quantify the dynamic process of food-seeking behavior in a "tube assay", which was modified from T-maze (Now Fig. 1h). As expected, in agreement with the food-patch assay, both hungry flies and the satiated flies with activated Taotie neurons walked toward and accumulated at the side with food odor, without being able to feed on the food (Fig. 1h,i).

It is interesting to see that the phenotypes evoked by long-term activation of Taotie neurons revert back to normal following cessation of the activation. It would be informative to see whether this reversion is due to a compensatory decrease in food intake below baseline levels; to test this, the authors could look at food intake, for example, 1 day following the cessation of neuronal activation.

We thank the reviewer for this interesting idea. We found that, in the reversion period, flies exhibited a compensatory decrease of food content below basal levels, suggesting that reversed obese phenotypes are the result of dramatically decreased food intake during this period (Supplementary Fig. 2b).

Figure 3d - the authors should perform a statistical comparison of the groups shown in this panel.

We added a statistical comparison of the groups in Figure 3d as suggested.

Page 8 - Previous studies have shown that flies use distinct behavioral and neuronal mechanisms to control feeding behavior in response to different periods of starvation. I would suggest that the authors could mention these previous studies, particularly Itskov et al, 2014 and Inagaki et al, 2012.

We apologize for the omission, and have now mentioned these studies and cited both Itskov et al, 2014 and Inagaki et al, 2012 in the revised version of our manuscript.

Page 8 - the authors write 'the difference between the activated group and the starved group became progressively smaller, suggesting that in the activated flies, the impact induced by *Taotie*-Gal4 activation was absorbed by the assumed hunger mechanism'. This description is unclear and not well explained. I would suggest modifying this sentence to read 'the difference between the activated group and the starved group became progressively smaller with increased starvation time, suggesting that *Taotie*-Gal4 activation mimicked the effect of 12h starvation'.

We thank the reviewer for this suggestions, we have now incorporated this sentence into the revised version of our manuscript.

Page 9 - 'After activation for 30 minutes or more in the absence of food, *Taotie*>*CsChrimson* flies exhibited strong food intake in the subsequent feeding assay, without further activation required (Fig. 6a,b).' It is not completely clear from this explanation that the activation is stopped before the assay begins; I suggest rewriting the sentence slightly to make this point clearer.

We agree, and have now rewritten this sentence to describe our experimental procedure more clearly: 'The satiated *Taotie*>*CsChrimson* flies first received light stimulation for a minimum of 30 minutes in the absence of food. The flies were then moved into darkness before the start of the subsequent feeding assay. Following this procedure, *Taotie*>*CsChrimson* flies exhibited strong food intake, with no further activation required (Fig. 5a and Supplementary Fig. 6a)'.

The only phenotype which does not fit the fact that the neuronal manipulation mimics starvation is the fact that the flies do not ingest bitter food. It has been clearly described that starvation makes flies not avoid bitter food anymore. The authors should put their findings in the context of these findings. Maybe the induced starvation state is not strong enough

to show that effect or the bitter content of food is too high?

We thank the reviewer for sharing this interesting observation. We found that, as shown in Supplementary Fig. 5g-i, activation of Taotie neurons evoked a 'mild' starvation state, which is different from intensively starved condition. Activation of Taotie neurons mimic the effect of 12 hours starvation. Although flies in both cases prefer palatable sugars (Fig. 4d,e and Supplementary Fig. 5a-f), both flies starved for 12 hours and the satiated flies with activated Taotie neurons avoided bitter food (Fig. 4d,e and Supplementary Fig. 5a-f). Only after starved for 36 hours, the 'severely' hungry wild type flies began to eat bitter food (Supplementary Fig. 5i). These results suggest that, as the reviewer pointed out in one of the earlier questions, 'flies use distinct behavioral and neuronal mechanisms to control feeding behavior in response to different periods of starvation'.

Page 9 - 'strongly argues against that these neurons locate in the input..' is poor grammar. I would suggest 'strongly suggests that these neurons are not located in the input or output pathways of feeding control, but have a central position...'.

We amended this sentence according to reviewer's suggestions.

The experiment in which the authors briefly activate the Taotie neurons followed by a delay period is very informative as to their temporal role. In this experiment, there is no food present during the delay period, which is an important factor. I would suggest that we could learn more about the sustained phenotype by also repeating the experiment with food present in the delay period. In this way, 1) the experiment could be extended to longer time points, which is currently limited by when the control group attains the same feeding level as the activated flies due to starvation; and b) it is possible that a transient activation activates a long-lasting motivational state that is only reversed when food is presented. If this is the case, then the presence of food during the delay period would abolish the persistent effect following activation. Either way, this experiment would give an interesting result.

We thank the reviewer for such an interesting experimental design. We found that providing food in the delay period effectively abolished the long-lasting motivational state, i.e. flies behaved as if satiated in the subsequent feeding test, suggesting that the motivational state of hunger is reversed by food-derived satiety (Supplementary Fig. 7). These data also lend strong support to our hypothesis that satiety counteracts the effects evoked by Taotie neuron activation or by hunger.

Page 9 - the subesophageal ganglion has been renamed as the subesophageal zone, according to Ito et al, 2014.

We thank the reviewer for pointing out this change in nomenclature, we have now corrected this

phrase accordingly.

Page 10 - the imaging result showing that these neurons are activated following starvation is very nice. However, in the description of the result, the authors should be more specific about what the data is. Rather than 'the PI region of hungry flies exhibited elevated reporter fluorescence, indicating that these neurons are highly activated when flies are hungry', they should specify that they are using a reporter of intracellular calcium levels, for example 'the neurons in the PI region exhibited increased GCaMP fluorescence following starvation, indicating that these neurons are activated, with high intracellular calcium levels, when flies are hungry'. Likewise, the authors should indicate that 'feeding hungry flies for 1 hour reduced the calcium level of these neurons to the level of satiated flies', rather than the 'activities of these neurons'.

We thank reviewer's assistance in expressing our findings more clearly. The imaging results have now been rewritten to use GCaMP signals in replacement of the activity of Taotie neurons.

The authors indicate that other neurotransmitters and neuropeptide systems are unlikely to interface with Taotie neurons based on their previous behavioural screen. To make this claim, the authors need to show the results of this behavioural screen. Also I am not sure that you can take these negative results as strong evidence. Especially not that this shows that the Taotie neurons are part of a high-level neuronal circuit. Absence of evidence is not evidence of absence.

We thank the reviewer for pointing this out. Indeed, we lack strong evidence that could rule out other neurotransmitters and neuropeptide systems for their interfacing with Taotie neurons. Our behavioral screen covered 400 GAL4 lines with neural expression, including dopamine, octopamine, serotonin, NPF and sNPF, none of these exhibited strong feeding phenotype as *Taotie-GAL4*. The purpose of the screen was to 'zoom in' onto the circuit that gives the most significant results, not a careful characterization of all GAL4 drivers we used. It is therefore possible that, upon careful inspection, activation of some GAL4s might also induce low levels of overfeeding behavior in satiated flies. From the experiments which narrowed down a functional subset of *Taotie-GAL4* neurons with various GAL80, we showed that the neurons with *Th-GAL80*, *Gad-GAL80* and *Cha-GAL80* and the corresponding dopaminergic, GABAergic and cholinergic systems are not required in Taotie neurons (Fig. 6b,d).

To investigate the interaction of Taotie neurons with other neurons in feeding process, we performed additional experiments of neural epistasis, and found that Taotie neurons interacted antagonistically with multiple satiety signals to coordinate feeding behavior (Supplementary Fig. 19). Additionally, in this revision, as suggested by the reviewers, we avoid the use of 'high-level'

neural circuit when discussing Taotie neurons.

Page 14 - the authors note that 'In the periphery, both olfactory and gustatory inputs related to feeding are affected by hunger'. The authors should cite references 17 and 18 in addition to 3 and 45 to support this point.

We thank the reviewer for this suggestion, all four references are now included in the revised version.

Page 16/17 - the authors suggest that 'as the serotonin neurons project broadly in the brain, they are presumably more suitable for modulating arousal globally'. I do not agree with this statement, and there is no evidence to suggest that the serotonergic neurons that evoke hunger are generally modulating arousal. Further, I do not quite understand the point the authors are trying to make here. In fact, it seems to me that the behavioural phenotypes of Taotie neurons are very similar to those demonstrated by Albin et al, 2015 for a subset of serotonergic neurons. Thus, these Taotie neurons are not the first demonstration of neurons in *Drosophila* that represent a motivational state of hunger and modulate different behaviours accordingly. However, this does not detract from the importance of this study but the authors should still discuss this work at the same level as theirs. Further, the authors mention that 'activating all serotonergic neurons with Tph-GAL4 in satiated flies did not elevate their feeding motivation'. This is true and was in fact shown by Albin et al, 2015. This indicates that global manipulations obscure the phenotypes observed upon manipulations of small subsets of neurons, and does not in any way invalidate the findings of Albin et al.

We agree with the reviewer. The original sentence on 'serotonin neurons ... are presumably more suitable for modulating arousal globally' was misleading. What we tried to express is that serotonin neurons, with their broad projections, are suitable for evoking a global arousal state (ie, hunger state) during hunger response to motivated animals to search food or feeding.

We thank the reviewer for all useful suggestions on how to rewrite this part of the discussion. In our previous discussion on motivational neurons on feeding, we found that it was difficult to interpret the results of serotonin neurons in terms of neural circuit, and this 'uncertainty' was reflected by our writing. Results from Albin et al, 2015 and from our own screen showed that manipulating the overall serotonergic system with *Tph-GAL4* did not produce over-feeding phenotype. Intriguingly, Albin et al, 2015 also showed that a subset of serotonin neurons is involved in the motivational aspects of feeding. Therefore, a simple hypothesis would be that the downstream signaling of serotonin neurons consists of at least two counteracting pathways with opposite effects, one promoting feeding behavior and the other inhibiting feeding behavior. There were no expression patterns of these responsible serotonin neurons (the location and the number of

Reviewer #3 (Remarks to the Author):

This manuscript addresses the fundamental question of energy balance and the control of feeding behavior in flies. A group of neurons is identified using the Taotie-gal4 driver line that promotes feeding behavior upon neural activation in satiated flies. Stimulation of these neurons using the Taotie driver induces an obese phenotype due to overfeeding, while their inhibition reduces feeding even after starvation. Taotie-activated flies present similar appetite as 12hr-starved animals, with identical food preference. Activation of Taotie neurons induces Dilp retention in insulin-producing cells, suggesting that the Taotie neurons could modulate feeding behavior through a general control of insulin signaling. Experiments are well crafted and in many instances support the conclusions of the paper. The message is significant and should contribute to a better knowledge of the links between homeostatic/physiological hunger and neurological activation of feeding. However, the manuscript does not address the crucial question of homeostatic signals upstream of the taotie neurons, despite the possibility of an obvious molecular link. In conclusion several major and minor remarks need to be addressed before publication.

We highly appreciate the reviewer's positive comments on the topic, approach and significance of our research. We were very interested in the upstream signals leading to the activation of Taotie neurons. The obvious and interesting candidate was, as the reviewer suggested, *Gr28b.b*. Briefly, our results with *Gr28b.b* null mutants and overexpression of *Gr28b.b* suggested that *Gr28b.b* is not required for hunger evoked events, not it is involved in feeding regulation in either starved or satiated flies (Supplementary Fig. 12). In the revision, we added additional experiments to investigate the upstream and downstream signals related to the activation of Taotie neuron. Detailed experiments are described below in the replies to Major comments.

Major comments:

1- The Taotie-Gal4 line contains the promoter region of the gustatory receptor *Gr28b.b* gene. This suggests that this gustatory receptor could participate in central nutrient sensing by the Taotie neurons (See Miyamoto et al. 2012 for the case of *Gr43a*). Why is this not experimentally tested, nor mentioned in the paper? This is a crucial point that needs to be addressed, since it could provide a key missing link between the nutritional status of the animal and the activity of the Taotie neurons.

We thank the reviewer for these constructive suggestions. *Gr28b.b* is an interesting molecule which, although classified as a gustatory receptor, plays multiple unique roles in the nervous system other than taste sensation. Therefore, we conducted behavioral tests in flies with either overexpressed full-length *Gr28b.b* or flies with null *Gr28b.b* mutant under both satiated and

starved conditions. However, these manipulations did not alter the amount of food intake significantly in our paradigm (Supplementary Fig. 12).

One previous study reported that the gustatory receptor *Gr43a* functions as an internal nutrient sensor in the brain and activation of Gr43a neurons causes an aversive sensation in satiated flies. Thus, we expect that co-activation of Taotie neurons and Gr43a neurons would generate opposite effects on feeding behavior. We found that the feeding phenotype evoked by activation of Taotie neurons were suppressed by co-activated Gr43a neurons (Supplementary Fig. 19c). In addition to Gr43a, we also co-activated Taotie neurons with other satiated signals, and found that overeating behavior induced by activation of Taotie neurons was reduced by activating several known satiated signals at the same time, suggesting antagonistic interactions between Taotie signals and the satiation signals (Supplementary Fig. 19a,b).

In order to establish a causal link between internal nutritional status of the animal and the activation of Taotie neurons during hunger response, we manipulated hemolymph glycemia via diet or drug treatment, and measured the calcium influx in Taotie neurons in responding to different treatments (Fig. 6g,h). Treatment with non-nutritious sugar, protein-rich food, or hypoglycemic drugs resulted in hypoglycemia, also lead to high activities in Taotie neurons, while all treatments of nutritional sugars that resulted in elevated glycemia, maintained with basal level activities in Taotie neurons (Fig. 6g,h), strongly arguing that neural activity of Taotie neurons were regulated by hemolymph glycemia levels. This conclusion is further confirmed by our re-feeding experiment that caloric sugars effectively suppressed neural activity of Taotie neurons, while non-caloric sugars and proteins-rich food did not inhibit neural activity of these neurons (Supplementary Fig. 14).

2- It is not clear why taotie neuron activation takes 30 min. to induce feeding. This seems desperately long for a circuitry that would transduce physiological hunger into neurological hunger in physiological conditions.

The reviewer addresses a very interesting point. Optogenetics has been shown to quickly activate target neurons at the cellular level. At the behavioral level, optogenetic manipulation can induce fast response (in seconds) in both sensory and motor circuits, at least in our hands. However, for activating motivational circuits with optogenetics, we expect the time scale would be different from our experience. Slow response is one indication to us that Taotie neurons is part of motivational circuits, other than sensory or motor circuits. Additionally, hunger in general is more likely a gradual procedure, not a rapid response, as the level of physiological indicators of hunger

progressively increase in physiological conditions.

We found that light-dependent optogenetic stimulation of Taotie neurons induced feeding behavior as fast as 15 minutes, although stimulation of 30 minutes produced more robust and stable results (Fig. 5a). More importantly, when forcibly activating Taotie neurons to evoke neurological hunger in satiated flies, the physiological satiation signals in these flies would counteract the optogenetic activation. The time for the body to resolve the conflict between neurological hunger versus physiological satiation to eventually initiate feeding action would reflect as a long delay.

3- There is a hole in the experimental logic describing the link between the Taotie neurons and the IPCs. In fed animals, Taotie neuron activation blocks insulin release. However, this cannot be causal to the observed feeding increase, since blocking IPC activity using Kir2 or Shits is not sufficient to induce feeding in these conditions (see Suppl. Fig. b,c).

We thank the reviewer for pointing out this apparent logical issue. Our results suggest that activation of Taotie neurons evokes at least two coordinated yet distinct events: activating neurological hunger pathway and blocking the inhibitory effects of satiation pathways. On the one hand, our results with suppressing insulin cells with Kir2.1 and Shibire indicated that blocking the satiation pathway alone in satiated flies did not result in more feeding (Supplementary Fig. 16b,c). This suggests that silencing the satiety signals alone is not sufficient to induce feeding behavior in satiated flies. On the other hand, activation of Taotie neurons inhibits the secretion of Dilp2 in these flies (Fig. 7a and Supplementary Fig. 17), suggesting that the activity of Taotie neurons may serve to keep the activities of IPC cells at a low level, since an otherwise high level of activity in IPC induces suppression of feeding (Supplementary Fig. 16a).

It is also possible that, while activating a single satiation pathway reduces feed motivation, we have to block all satiation pathways to release their inhibition in order to promote feeding motivation. Our neural epistatic experiments showed that forcibly activating Dilp2 cells and other satiety signals reduced but not fully suppressed the overeating behavior that was evoked by activation of Taotie neurons, therefore activation of Taotie neurons exhibits broad inhibition effects on multiple satiation pathways (Supplementary Fig. 19). However, as shown in Supplementary Fig. 16b,c, without Taotie neuron activation, inactivation of only one satiation pathway with Kir2.1 or Shibire on the IPC cells, is not sufficient to overcome the suppressive actions of the other satiation signals to evoke hunger sensation and feeding. In addition, these results further suggested that Taotie neurons intertwining antagonize with satiety signals broadly, and in turn demonstrated that

multiple downstream targets of Taotie neurons coordinate status of physiological and neurological to govern feeding behavior.

To address these points better, we have now added more discussions in the revised version of our manuscript.

Minor comments:

In many instances, redundant data and/or control experiments are presented in independent panels, which should be moved to suppl. Material. For instance: Fig. 1a,b,e,f,j; Fig. 4b,d,h; Fig. 6b,e.

We agree with the reviewer, and have now re-organized our data and panels (We removed the entire Fig. 4 into Supplementary Fig. 4, and Fig. 6b,e to Supplementary Fig. 6a,b) according to the suggestions from several reviewers.

2- Why using colorimetric quantifications of a food dye on whole flies where a cafe assay would be more quantitative?

We chose this colorimetric method for quantification primarily to conduct the behavioral screen. Comparing to the CAFE assay, colorimetric quantification is more stable, at least in our hands, and is better-suited for high-throughput analysis. Moreover, we found that for the CAFE assay, flies have to first learn to feed from the food filling capillary, thus adding extra layers of complexity to the experimental procedure to analyze feeding behavior.

3- Interpretation of results in Fig. 1d seems misleading: starvation enhances the effect of taotie>trpA1 by three fold, meaning that taotie neuron activation is only part of the nutritional response, but not sufficient for full response.

We thank the reviewer for this careful observation. In the panel addressed (now Supplementary Fig. 1a), the hungry flies exhibited greater food ingestion when starved for 48 hours. In additional experiments, we compared several features of feeding behavior in flies with activation of Taotie neurons with those in hungry flies. The amount of food intake and food preference in these flies were identical to 'mildly' starved flies (12 hour starvation) (Fig. 4 and Supplementary Fig. 5), which is distinct from 'severely' starved flies (more than 36 hour starvation) (Fig. 4b and Supplementary Fig. 5g-i). Thus, at least for 12 hour starvation, Taotie neuron activation is sufficient for full response.

4- A convincing test that activation of taotie neurons increase food search behavior would be to look for general increase of motility associated to food search. Are animals more motile upon taotie neuron activation in the absence of food?

We have now added a new paradigm to quantify food search behavior in the revised version of our manuscript. To quantify the portion of hungry flies walking toward food odor in the absence of food,

we set up a modified T-maze. In contrast to control satiated flies, the satiated flies with activated Taotie neurons exhibited similar tendency to walk toward food odor as the hungry flies (Fig. 1h,i), suggestion that activating Taotie neuron evokes food searching in satiated flies.

Previous studies indicated starvation induces hyperactivity in flies. We asked whether activating Taotie neurons causes an increase of locomotion. We analyzed the movement of flies in an open field without food, but did not find significant changes of speed upon Taotie neurons activation. The reason for the discrepancy is currently not clear to us.

5- In the absence of more data concerning the effect of Taotie neuron activation on obesity, it seems useless to compare this with the obese syndrome in human and its reversibility. Are Taotie-activated obese flies presenting some characteristics of the obesity syndrome (insulin resistance, general inflammatory response etc.)?

We thank the reviewer for pointing this out. We carefully examined the expression of known insulin pathway genes upon activation of Taotie neurons, and added the results to our revised version. The mRNA levels of *eIF4E* and *S6K*, two targets of the insulin pathway, were found to be reduced following activation of Taotie neurons, whereas the mRNA levels of *4E-BP*, a negative regulator of the insulin pathway, increased (Supplementary Fig. 3b). Interestingly, mRNA levels of *Dilp2*, *Dilp3* and *Dilp5* were also dramatically increased (Supplementary Fig. 3a). Thus, expression results show that, although mRNA levels of *Dilps* are up-regulated, the downstream insulin signaling is reduced with activation of Taotie neurons, suggesting an impaired insulin signaling upon activation of Taotie neurons, a key feature of insulin resistance in human obesity.

We also checked inflammatory responses in obese flies induced by Taotie neuron activation. The mRNA levels of *TNF* and *Upd3* (cytokines in flies) were dramatically increased, suggesting an elevated general inflammatory response in these obese flies (Supplementary Fig. 3c). Intriguingly, this resembles the obese phenotype in humans, which is commonly associated with low-level chronic inflammation. Therefore, our results suggest that activity of Taotie neurons is crucial for physiological homeostasis. Thus, we strongly believe that manipulating Taotie neurons activities leads to obese phenotypes with certain characteristics of obesity in human.

6- In fig. 6d, the long-lasting motivation for feeding induced by taotie neuron activation could simply be due to the non-physiological activation procedure. Therefore, the proposed conclusion seems inappropriate.

We thank the reviewer for pointing this out; however, we respectfully disagree with this conclusion. Optogenetic manipulation is a useful, although not physiological, activation procedure. With this

approach, we can bypass the physiological status or neurological context to manipulate target neurons and observe the behavioral consequence. Our results on optogenetic activation of gustatory neurons and motor neurons indicated that these manipulation is still physiological relevant, ie, these neurons exhibit fast-on and fast-off response, and do not show long-lasting effects (Fig. 5b). Those neurons would serve as comparable controls to the Taotie neurons which showed long-lasting effects by similar treatment in our paradigm. Moreover, we found that providing flies with food in the delay period suppressed the long-lasting motivational state (Supplementary Fig. 7), suggesting the long-lasting motivation for feeding by Taotie neuron activation is still responsive to physiologically relevant events. Besides optogenetics, *Taotie>TrpA1* flies also generated long-lasting after effect (Supplementary Fig. 6c-f). Therefore, the long-lasting effects by activating Taotie neurons is likely a unique feature of the Taotie neurons or its circuit, rather than an artificial result without physiological relevance.

7- The authors should avoid emphatic terms like "master feeding center".

We agree with the reviewer, and have now amended our revised manuscript accordingly.

8- Fig 7 should present a magnification of the PI cluster showing individual neurons for clarity.

In the revised manuscript, we have included stacks and single sections of two color co-labeling brain images, to show individual neurons of Taotie neurons and Dilp2 cells in the PI region (Supplementary Fig. 15a-c). We also provided a video of 3D reconstruction of individual neurons in the PI region to visualize the spatial relationship of Taotie neurons and Dilp2 cells (Supplementary video 1).

9- The authors should comply to the nomenclature, i.e. insulin-producing cells are called IPCs, not DPMNs.

We amended the names to IPCs according to the conventions of current nomenclature.

REVIEWERS' COMMENTS:

Reviewer #2 (Remarks to the Author):

Review of Identification of a feeding circuit for physiological homeostasis in *Drosophila*

As previously stated, this study provides an important advance in our understanding of the neuronal control of feeding behaviour and physiological responses to starvation. The paper is well-constructed and the experiments very thorough.

Following the review process, we are satisfied that the authors have answered all of the points raised by the reviewers, and can therefore recommend this paper for publication in *Nature Communications*. We actually applaud the effort the authors put into revising the manuscript. We are also happy that our suggestions were of use to the authors. However, there remain a few minor points that the authors should address in the manuscript, however, prior to publication. We strongly suggest that the authors remove the word circuit from the title. They actually characterize one set of neurons and that does definitely not count as a circuit.

Throughout the manuscript and the comment to the referees the authors refer to the argument that they have screened lines representing different types of neurons and molecules which were negative. To be able to make that statement the authors should show the data of the screen. That would also add significant value for the community. It will stop people from retesting the same lines over and over again.

The authors decided not to introduce the *elav-Gal80* data at the beginning of the manuscript. Then they have to restrain themselves from writing in the text that they are testing *Taotie* NEURONS for an effect. They can only explicitly say that after they have shown it using *elav-Gal80*. Until that stage in the narrative the effects they see can be due to manipulations of other cell types. Either they mention the *elav-Gal80* data early or do not use the word neuron until that experiment is done.

We still do not agree that the authors demonstrate that their data show that the neuronal manipulations induce ACTIVE food searching. The effects they show can also be caused by the animals walking around randomly and stopping at the location of the odour. Their conclusion is not supported by the evidence. For that they would need to do a dynamic analysis of movement direction. What the authors show is a preference readout. At this stage they have to use that terminology or something similar.

All figures: It is still not clear what type of test was used in which figure panel for which experiment and what is exactly compared. The authors have to clarify this better.

All figures: All legends state that error bars indicate SEM, but many panels show box and whisker plots. Could the authors clarify whether these plots actually show SEM, or if not, clarify what the box and whiskers represent in the figure legend.

Figure 1c: We do not understand exactly how the statistics were done. Did the authors use a 4x2 Fisher's exact test? Did they compare all 3 groups, or compare each control with the test group separately?

Figure 2: We would recommend changing the x-axis label to 'Days' - 'Days on stimulation' is misleading since they are only stimulated for the first 4 days.

Figure legend 6a: There is a typo – GFP is misspelt as 'GPF'.

Figure legend 6: We would recommend the authors add "differences to the satiated group" to the sentence explaining what the letters in g and h represent.

Introduction: "The motivational aspect of feeding, influenced by food quality and hungry state, play critical role in overfeeding which leads to overweight and obesity. Despite this is" – this part has grammatical issues.

Page 5: "like the starved flies, these flies did not exhibit spontaneous PER in the absence of food". I believe this to be true, particularly for flies starved for <12 hours. However, Yong et al (2016) do state that starved flies do exhibit spontaneous PER. Do the authors have an explanation to explain this discrepancy?

Page 6: "a reduction of food intake after cessation of light stimulation" – We would emphasise that the food intake is reduced below baseline levels.

Page 7: "hemostasis" – typo?

Page 7: “the activity levels of Taotie neurons function as set points to fine tune the energy homeostasis” – We disagree with this interpretation. It is possible that Taotie neurons control the body weight set point, but since food intake remains equally elevated after 4 days of stimulation, I find it more likely that the Taotie neurons in fact induce an artificial “error” signal, telling the fly that it has deviated from the normal set point – particularly as the neurons’ activity correlates with hemolymph glycemia, as would be expected for neurons calculating an error signal. In any case, we would not claim that these neurons control any negative feedback set point.

Page 7: “to selectively inhibit the activities of targeted neurons” – TNT inhibits synaptic output, not neuronal activity

Page 10: “focusing a laser beam on the head, instead of the thorax, was sufficient to evoke PER behavior” – upon reading this we thought that it contradicted the finding that activating Taotie neurons would not induce spontaneous PER in the absence of food, but upon reading the methods we realised that this experiment was performed in the presence of a food stimulus. We think that this should be more clear in the main text – eg. “was sufficient to evoke PER behaviour in response to sucrose”.

Discussion: “blind intervention into general physiological or neural processes related to feeding will presumably be less effective than targeting the underlying causes for obesity” – we understand the point here – that a targeted approach to treating the causes of obesity is important – but we think it could be written a little more clearly. We also think that it is a bit overstated. The time-scale of the intervention in the paper and the one which will make a difference in a patient are very different.

“neurons whose activities represent set-points for homeostasis” – again we do not believe this is true. If so, the authors should at least point out exactly what they are a set point for, rather than just general “homeostasis”.

“persisting (as physiological states) or transient (instantaneous result from processing physiological signals)” – we do not understand the point here. We realise that transient activation of Taotie neurons evokes a persistent hunger state, but this is not quite what the sentence is asking. “elevated appetite is a consequence of evoking an endogenous neural response toward hunger” – We don’t understand this sentence exactly.. Could the authors clarify?

We think it would be pertinent for the authors to also reference two recent papers that came out while the paper was in review:

Chen et al, *eLife*, 2016 – show that transient activation of *Agrp* neurons evokes persistent hunger state. Another similarity between *Agrp* neurons and Taotie neurons.

Jourjine et al, *Cell*, 2016 – identify another set of neurons involved in starvation responses in the *Drosophila* brain.

In figure 1c, the abdomen scores are notated as 1-4, whereas in the Materials & Methods, they are called 0-3. These should be made consistent.

We could not find an exact description of the diets on which flies were pre-fed for figures 6g-h (ie. The concentrations of the sugars/drugs given). This should be added to the materials & methods section.

WE would like to reiterate that apart from these minor points, we are very impressed with the quality and scope of this manuscript, and would wholeheartedly recommend it for publication in *Nature Communications* once these concerns have been addressed.

Reviewer #3 (Remarks to the Author):

The authors have sensibly improved their manuscript by providing important supplemental data as requested and satisfactory responses to all my previous remarks.

Author response:

We wish to thank the editor and reviewers for the efforts they have put into reviewing and improving our work. We revised our manuscript and addressed the new questions from the reviewer. Please find our point-by-point response to the reviewer's comments in the following text.

Reviewer #2 (Remarks to the Author):

As previously stated, this study provides an important advance in our understanding of the neuronal control of feeding behaviour and physiological responses to starvation. The paper is well-constructed and the experiments very thorough. Following the review process, we are satisfied that the authors have answered all of the points raised by the reviewers, and can therefore recommend this paper for publication in Nature Communications. We actually applaud the effort the authors put into revising the manuscript. We are also happy that our suggestions were of use to the authors. However, there remain a few minor points that the authors should address in the manuscript, however, prior to publication.

We thank the affirmation of our work from reviewer#2. In addition, we highly appreciate the helpful suggestions by reviewer #2.

We strongly suggest that the authors remove the word circuit from the title. They actually characterize one set of neurons and that does definitely not count as a circuit.

In our revised version, we have removed the word circuit from the title. The title is changed into 'Taotie neurons regulate appetite in *Drosophila*'. However, we would like to point out that Taotie neurons are not the only neurons studied in this research. In the efforts to dissect the related neural circuit, we found that Taotie neurons regulate Dilp cells, establishing a hierarchical circuit relationship.

Throughout the manuscript and the comment to the referees the authors refer to the argument that they have screened lines representing different types of neurons and molecules which were negative. To be able to make that statement the authors should show the data of the screen. That would also add significant value for the community. It will stop people from retesting the same lines over and over again.

First of all, we promise that all our data including the list of GAL4 lines screened are available upon request.

We have been conducted a screen of GAL4 lines including dopamine, octopamine, serotonin,

Dilp2, Akh, Dsk, NPF and sNPF. None of these exhibited as strong feeding phenotypes as *Taotie-GAL4* after activation. Consistent with previous study, these neurotransmitters and neuropeptides have been well-known that participate in subprogram of feeding behavior, and only have slightly effects on regulating food intake (*Inagaki et al, Cell, 2012, Marella et al, Neuron, 2012, Albin et al, Current Biology, 2015, Yang et al, PNAS, 2015, Yu et al, Elife, 2016* and reviewed in *Pool and Scott, Current Opinion Neurobiology, 2014*).

As for the other lines, we mentioned in the previous response letter that the primary purpose of the screen was to quickly 'zoom in' onto the circuit with the most significant results, instead of a careful characterization of all GAL4 drivers that we collected from different labs over the years. As a reviewer suggested last round, 'absence of evidence is not evidence of absence'. We believe it is counter-productive to include the data with the false negative rate yet estimated in this publication. We currently conduct additional studies to carefully inspect neurons labeled by these GAL4 lines in different feeding behavior paradigms.

Not including the whole list of screened lines is also consistent with the practice of the field. For example, the screens conducted by these following studies recently did not include the full lists of lines: *Manzo et al. in PNAS (2012), Marella et al. in Neuron (2012), Flood et al. in Nature (2013), Mann et al. in Neuron (2013), Pool et al. in Neuron (2014), Albin et al. in Current Biology (2015)* and *Yapici et al. in Cell (2016)*.

The authors decided not to introduce the elav-Gal80 data at the beginning of the manuscript. Then they have to restrain themselves from writing in the text that they are testing Taotie NEURONS for an effect. They can only explicitly say that after they have shown it using elav-Gal80. Until that stage in the narrative the effects they see can be due to manipulations of other cell types. Either they mention the elav-Gal80 data early or do not use the word neuron until that experiment is done.

At the beginning of our manuscript, we used neural activators via thermogenetic and optogenetic approaches -- both techniques are typically suitable to activate neurons rather than other cell types. The collection of GAL4 lines that we screened through were previously studied by others labs for the corresponding neurons they labeled. It seems satisfactory to conclude that the Taotie neurons are responsible for the feeding phenotypes before further characterization.

However, we concerned that combining the neuron activation techniques with the neuron labeling

lines still would not guarantee that the Taotie cells which we activated are indeed neurons. Therefore we conducted the elav-GAL80 experiments later on to resolve the issue. Our approach is similar with previous studies (*Gordon and Scott*, Neuron 2009, *Mann et al.*, Neuron 2013, *pool et al.*, Neuron 2014 and *Hopper et al.*, Elife 2015).

We still do not agree that the authors demonstrate that their data show that the neuronal manipulations induce ACTIVE food searching. The effects they show can also be caused by the animals walking around randomly and stopping at the location of the odour. Their conclusion is not supported by the evidence. For that they would need to do a dynamic analysis of movement direction. What the authors show is a preference readout. At this stage they have to use that terminology or something similar.

We thank the reviewer for raising this point. In the original manuscript, we showed that hungry flies (and Taotie activated flies) accumulated at the area containing sugars, suggesting an elevation of hungry-driven gustatory-based response toward potential food source. In the revised manuscript from last round, we showed that hungry flies (and Taotie activated flies) accumulated at the area containing food odors, even when no feeding was involved. The food odor induced accumulation happened rapidly (within 10 minutes), therefore we consider it an active process. Limited by the setup, we did not conduct single fly tracking for dynamic analysis of their movement.

Our results agreed with previous literatures (*Root et al.*, Cell (2011), *Park et al.*, Current Biology (2016) and *Min et al.*, Current Biology (2016)): starved flies are attracted to food odor in a dish and T-maze assay. The satiated flies with activated Taotie neurons exhibited similar attractive tendency to food odor, behave like the hungry flies in dish assay and in T-maze assay. Accordingly, we proposed that T-maze assay could reflect hunger or satiety states in flies and manipulation Taotie neurons mediated endogenous hunger-driven olfactory response.

Because the reviewer#2 was strongly against the use of 'active food searching', we now called the behavior response 'hunger-driven food odor attraction' in this revision.

All figures: It is still not clear what type of test was used in which figure panel for which experiment and what is exactly compared. The authors have to clarify this better.

We apologize for the ambiguity and made some changes in the revision. Considering the limitation on the number of words in figure legends, we used standard description of Statistics in each figure legends as well as in Materials & methods, and had additional descriptions for other specific

comparisons and analysis. We also used horizontal lines in each panel to indicate the experimental groups to be compared.

All figures: All legends state that error bars indicate SEM, but many panels show box and whisker plots. Could the authors clarify whether these plots actually show SEM, or if not, clarify what the box and whiskers represent in the figure legend.

We are sorry for the misunderstanding.

No error bars nor SEM were used or showed for the box and whisker plots.

For the panels with box and whisker plots, the whiskers label the minimum and maximum of the data set, the box includes the 25th to 75th percentile data points, and the line within indicates the median.

We amended the figure legends in revised manuscript to clarify the issue.

Figure 1c: We do not understand exactly how the statistics were done. Did the authors use a 4x2 Fisher's exact test? Did they compare all 3 groups, or compare each control with the test group separately?

We performed the *Fisher's exact* test to compare all three groups at different temperatures. In the revision, we added this information in the figure legend of Figure 1c.

Figure 2: We would recommend changing the x-axis label to 'Days'-'Days on stimulation' is misleading since they are only stimulated for the first 4 days.

We thank the reviewer for this thoughtful suggestion. We amended the revised manuscript accordingly.

Figure legend 6a: There is a typo-GFP is misspelt as 'GPF'.

We thank the reviewer to point the out. It was corrected in the revision.

Figure legend 6: We would recommend the authors add 'differences to the satiated group' to the sentence explaining what the letters in g and h represent.

We agree with the reviewer and amended the figure legend accordingly.

Introduction: 'The motivational aspect of feeding, influenced by food quality and hungry state, play critical role in overfeeding which leads to overweight and obesity. Despite this is'-this part has grammatical issues.

We now changed this sentence into 'The motivational feeding plays a critical role in overfeeding that leads to overweight and obesity. '

Page 5: 'like the starved flies, these flies did not exhibit spontaneous PER in the absence of food'. I believe this to be true, particularly for flies starved for <12 hours. However, Yong et al (2016) do state that starved flies do exhibit spontaneous PER. Do the authors have an explanation to explain this discrepancy?

Consistent with our findings, previous studies (*Pool et al.*, *Neuron* (2014) and *Flood et al.*, *Nature* (2013)) showed that food deprived flies do not exhibit spontaneous PER, but activated motor command neurons induce PER response even in the absence of food. We read the work of *Yong et al.* published on *Current Biology* (2016), but did not find the data to show that starved flies exhibit spontaneous PER.

During our research, we noticed that the time of day to perform experiments, the culture states of the flies, and the length of food deprivation influence PER activity in flies. Slight differences in the experimental protocol might generate very different phenotypes. We believe that maintaining a consistent condition is the key for obtaining consistent behavioral results.

Page 6: 'a reduction of food intake after cessation of light stimulation'-We would emphasise that the food intake is reduced below baseline levels.

We thank the reviewer for this suggestion, and amended the revised manuscript accordingly.

Page 7: 'hemostasis'-typo?

We thank the reviewer for careful inspection, and corrected the typo in the revision.

Page 7: 'the activity levels of Taotie neurons function as set points to fine tune the energy homeostasis'-We disagree with this interpretation. It is possible that Taotie neurons control the body weight set point, but since food intake remains equally elevated after 4 days of stimulation, I find it more likely that the Taotie neurons in fact induce an artificial 'error' signal, telling the fly that it has deviated from the normal set point-particularly as the neurons' activity correlates with hemolymph glycemia, as would be expected for neurons calculating an error signal. In any case, we would not claim that these neurons control any negative feedback set point.

We thank the reviewer to explain the alternative possibility clearly, which agrees with our working hypothesis. We do not have data to support that Taotie neurons control directly the body weight set point. We proposed that Taotie neurons control the set point of energy homeostasis, by acting both as a sensor and an activator. The exact mechanism of how the set point is adjusted by either the physiological state or a mismatch (or error signal) between current neurological state and

physiological state remains to be determined. It is possible that the Taotie neurons generate the error signal. Furthermore, the mechanism of how the set point is used to adjust the feeding behavior in response of hunger or the mismatch between neurological and physiological states is also not clear.

The fact that food intake remains equally elevated after 4 days of stimulation suggests that forcibly 'clamping' the Taotie neurons at high activity states (via optogenetics) still forces the downstream mechanism to work continuously to induce high feeding motivation, while ignoring the negative feedback (or error signals) from the physiological states for the past four days. Indeed, the suppressive signals are induced and are very powerful -- when the activity of Taotie neurons was allowed to free-run after the fourth day, the amount of food intake decreased to be lower than the baseline level.

Our data in Supplementary Fig. 2e and 2f showed that forcibly clamping the activity of Taotie neurons at different levels, which generates a condition that is resistant to negative feedbacks from physiological state, results in different degrees of obesity, further supporting the role of Taotie neurons on controlling (or adjusting) the set point.

Page 7: 'to selectively inhibit the activities of targeted neurons'-TNT inhibits synaptic output, not neuronal activity.

We agree with the reviewer, amended the sentence accordingly in revision.

Page 10: 'focusing a laser beam on the head, instead of the thorax, was sufficient to evoke PER behavior'-upon reading this we thought that it contradicted the finding that activating Taotie neurons would not induce spontaneous PER in the absence of food, but upon reading the methods we realised that this experiment was performed in the presence of a food stimulus. We think that this should be more clear in the main text-eg. 'was sufficient to evoke PER behaviour in response to sucrose'.

We thank the reviewer's helpful suggestion and now change this sentence accordingly.

Discussion: 'blind intervention into general physiological or neural processes related to feeding will presumably be less effective than targeting the underlying causes for obesity'-we understand the point here-that a targeted approach to treating the causes of obesity is important-but we think it could be written a little more clearly. We also think that it is a bit overstated. The time-scale of the intervention in the paper and the one which will make a difference in a patient are very

different.

We thank the reviewer for this suggestion, and amended the manuscript accordingly.

We acknowledge that the time scales for similar biological processes between different organisms, for example, the lifespan in human and *Drosophila*, are very different. How to calibrate or compensate for the temporal variations to achieve a unified description of the same biological process across different species is far from clear. However, it seems, at least in some cases of neurodegeneration, mutant flies show phenotypes in weeks while it takes tens of years for the same phenotypes to develop in human. Discussion of time-scale variation, although fascinating, is beyond the topic of this paper and the expertise of the authors.

'neurons whose activities represent set-points for homeostasis'-again we do not believe this is true. If so, the authors should at least point out exactly what they are a set point for, rather than just general 'homeostasis'.

Our data in Supplementary Fig. 2e and 2f showed that forcibly clamping the activity of Taotie neurons at different levels results in different degrees of obesity, that is, the activity levels of Taotie neurons are directly correlated with the body weight. Considering the context established by data from other experiments in this work, the most straightforward conclusion derived from this result is that the activities of Taotie neurons represent set-points of feeding motivation or hunger sensation.

We thanks the reviewer for pointing out his/her concerns. In the revised manuscript, we rewrote this sentence into 'we showed that modulating the activity level of Taotie neurons fine-tunes the body weight'.

'persisting (as physiological states) or transient (instantaneous result from processing physiological signals'-we do not understand the point here. We realise that transient activation of Taotie neurons evokes a persist hunger state, but this is not quite what the sentence is asking.

We thank the reviewer for pointing this out. Typically physiological states, such as hunger and fatigue, arrive slow and last for a long time (for hours), comparing with the action potentials of a neuron (with a time-scale of milliseconds). When physiological states are converted to neuronal signals and converge to a neuron for computation, it is reasonable to expect that this computation is also finished within milliseconds. Moreover, for optogenetics, the opening rate and close rate of channel-rhodopsin and its variants upon light stimulation are also within millisecond range,

thereby our optogenetic manipulation on neuronal activities is also rapid.

It is not known whether the neuron and related circuit that compute physiological states keep their calculation result transiently, just long enough to issue motor commands, or they hold the result for much longer to a comparable time-scale for physiological states. In term of hunger, our questions is whether neurological hunger (computing from physiological hunger) is transient (as a fast event) or persisting (as a condition or state). We found that transient activation of Taotie neurons evokes a persisting hunger motivation, suggesting that neurological hunger is not a transient event but a state lasting for hours.

We amended the sentence to make our question clear.

'elevated appetite is a consequence of evoking an endogenous neural response toward hunger'-We don't understand this sentence exactly. Could the authors clarify?

Here, we proposed that activation of Taotie neurons induces flies into an endogenous hunger state. This neurological hunger state triggers a series of behavioral changes, for example, being more attracted by food odors, higher tendency of PER to food source, higher quantity of food intake. In a fly being physiological satiated, these behaviors would be characterized as an apparent elevated appetite.

We think it would be pertinent for the authors to also reference two recent papers that came out while the paper was in review:

Chen et al, eLife, 2016-show that transient activation of Agrp neurons evokes persistent hunger state. Another similarity between Agrp neurons and Taotie neurons.

Jourjine et al, Cell, 2016-identify another set of neurons involved in starvation responses in the Drosophila brain.

We are grateful for the reviewer's suggestion, and included these papers in the discussion.

In figure 1c, the abdomen scores are notated as 1-4, whereas in the Materials & Methods, they are called 0-3. These should be made consistent.

We are sorry for the mistake in the figure legend. In the revision, the abdomen scores are indicated as 0-3 in both figure 1c and Materials & Methods.

We could not find an exact description of the diets on which flies were pre-fed for figures 6g-h (ie. The concentrations of the sugars/drugs given). This should be added to the materials & methods section.

We thank the reviewer for pointing this out. We added the description of the diets for calcium imaging experiments (Fig. 6g and 6h) in Materials & Methods.